# Crowd control: Reducing individual estimation bias by sharing biased social information

**Bertrand Jayles**[1,2]*, **Clément Sire**[3], **Ralf H. J. M. Kurvers**[1]

**1** Center for Adaptive Rationality, Max Planck Institute for Human Development, Berlin, Germany, **2** Institute of Catastrophe Risk Management, Nanyang Technological University, Singapore, Republic of Singapore, **3** Laboratoire de Physique Théorique, Centre National de la Recherche Scientifique (CNRS), Université de Toulouse – Paul Sabatier (UPS), Toulouse, France

* bertrand.jayles@ntu.edu.sg

**Data Availability Statement:** The data supporting the findings of this study are available at figshare: https://doi.org/10.6084/m9.figshare.12472034.v2.

**Funding:** B.J. and R.K. were partly funded by the Deutsche Forschungsgemeinschaft (DFG, German

## Abstract

Cognitive biases are widespread in humans and animals alike, and can sometimes be reinforced by social interactions. One prime bias in judgment and decision-making is the human tendency to underestimate large quantities. Previous research on social influence in estimation tasks has generally focused on the impact of single estimates on individual and collective accuracy, showing that randomly sharing estimates does not reduce the underestimation bias. Here, we test a method of social information sharing that exploits the known relationship between the true value and the level of underestimation, and study if it can counteract the underestimation bias. We performed estimation experiments in which participants had to estimate a series of quantities twice, before and after receiving estimates from one or several group members. Our purpose was threefold: to study (i) whether restructuring the sharing of social information can reduce the underestimation bias, (ii) how the number of estimates received affects the sensitivity to social influence and estimation accuracy, and (iii) the mechanisms underlying the integration of multiple estimates. Our restructuring of social interactions successfully countered the underestimation bias. Moreover, we find that sharing more than one estimate also reduces the underestimation bias. Underlying our results are a human tendency to herd, to trust larger estimates than one's own more than smaller estimates, and to follow disparate social information less. Using a computational modeling approach, we demonstrate that these effects are indeed key to explain the experimental results. Overall, our results show that existing knowledge on biases can be used to dampen their negative effects and boost judgment accuracy, paving the way for combating other cognitive biases threatening collective systems.

## Author summary

Humans and animals are subject to a variety of cognitive biases that hamper the quality of their judgments. We study the possibility to attenuate such biases, by strategically selecting the pieces of social information to share in human groups. We focus on the underestimation bias, a tendency to underestimate large quantities. In estimation experiments, participants were asked to estimate quantities before and after receiving estimates from other

Research Foundation) under Germany's Excellence Strategy – EXC 2002/1 "Science of Intelligence" – project number 390523135. The funders had no role in study design, data collection and analysis, decision to publish, or preparation of the manuscript.

**Competing interests:** The authors have declared that no competing interests exist.

group members. We varied the number of shared estimates, and their selection method. Our results show that it is indeed possible to counter the underestimation bias, by exposing participants to estimates that tend to overestimate the group median. Subjects followed the social information more when (i) it was further away from their own estimate, (ii) the pieces of social information showed a high agreement, and (iii) it was on average higher than their own estimate. We introduce a model highlighting the core role of these effects in explaining the observed patterns of social influence and estimation accuracy. The model reproduces all the main experimental patterns well. The success of our method paves the way for testing similar interventions in different social systems to impede other cognitive biases.

## Introduction

Human and non-human animal judgments and decisions are characterized by a plethora of cognitive biases, i.e., deviations from assumed rationality in judgment [1]. Biases at the individual level can have negative consequences at the collective level. For instance, Mahmoodi et al. showed that the human tendency to give equal weight to the opinions of individuals (equality bias) leads to suboptimal collective decisions when groups harbor individuals with different competences [2]. Understanding the role of cognitive biases in collective systems is becoming increasingly important in modern digital societies.

The recent advent and soar of information technology has substantially altered human interactions, in particular how social information is shared and processed: people share content and opinions with thousands of contacts on social networks such as Facebook and Twitter [3–5], and rate and comment on sellers and products on websites like Amazon, TripAdvisor, and Airbnb [6–8]. While this new age of social information exchange carries vast potential for enhanced collaborative work [9] and collective intelligence [10–13], it also bears the risks of amplifying existing biases. For instance, the tendency to favor interactions with like-minded people (in-group bias [14]) is reinforced by recommender systems, enhancing the emergence of echo chambers [15] and filter bubbles [16] which, in turn, further increases the risk of opinion polarization. Given the importance of the role of biases in social systems, it is important to develop strategies that can reduce their detrimental impact on judgments and decisions in social information sharing contexts.

One promising, yet hitherto untested, strategy to reduce the detrimental impact of biases is to use prior knowledge on these biases when designing the structure of social interactions. Here, we will test whether such a strategy can be employed to reduce the negative effects of a specific bias on individual and collective judgments in human groups. We use the framework of estimation tasks, which are well-suited to quantitative studies on social interactions [17–20], and focus on the *underestimation bias*. The underestimation bias is a well-documented human tendency to underestimate large quantities in estimation tasks [20–30]. The underestimation bias has been reported across various tasks, including in estimations of numerosity, population sizes of cities, pricing, astronomical or geological events, and risk judgment [20, 27–30]. To illustrate with one example, in [23] subjects had to estimate the number of dots (varying from 36 to 1010) on paper sheets. Subjects underestimated the actual number of dots in all cases. This study (and others) suggested that the tendency to underestimate large quantities could stem from an internal compression of perceived stimuli [23–25]. The seminal study by Lorenz et al. (2011) has shown that the effects of the underestimation bias could be amplified after social interactions in human groups, and deteriorate judgment accuracy [19].

In the present work, we investigate the effects of different interaction structures, aimed at counteracting the underestimation bias, on individual and collective accuracy (details are given below). Moreover, we investigate how these structures interact with the number of shared estimates in shaping social influence and accuracy. Previous research on estimation tasks has largely overlooked both of these factors. Thus far, research on estimation tasks mostly discussed the beneficial or detrimental effects of social influence on group performance [19, 31–37]. Moreover, most previous studies focused on the impact of a *single* piece of social information (one estimate or the average of several estimates), or did not systematically vary their number. In addition, in most studies, subjects received social information from *randomly selected* individuals (either group members, or participants from former experiments) [17–20, 32, 36–40]. In contrast to these previous works, in many daily choices under social influence, one generally considers not only one, but several sources of social information, and these sources are rarely chosen randomly [41]. Even when not actively selecting information sources, one routinely experiences recommended content (e.g., books on Amazon, movies on Netflix, or videos on YouTube) generated by algorithms which incorporate our "tastes" (i.e., previous choices) and that of (similar) others [42].

Following these observations, we confronted groups with a series of estimation tasks, in which individuals first estimated miscellaneous (large) quantities, and then re-evaluated their estimates after receiving a varying number of estimates $\tau$ ($\tau$ = 1, 3, 5, 7, 9, and 11) from other group members. Crucially, the shared estimates were selected in three different manners:

- *Random* treatment: subjects received personal estimates from $\tau$ random other group members. Previous research showed that when individuals in groups receive single, randomly selected estimates, individual accuracy improves because estimates converge, but collective accuracy does not [19, 20]. We hence expected to also find improvements in individual accuracy, but not in collective accuracy, at $\tau$ = 1. Furthermore, we expected individual and collective accuracy to increase with the number of shared estimates, as we anticipated subjects to use the social information better with an increasing number of shared estimates [43–45].

- *Median* treatment: subjects received as social information the $\tau$ estimates from other subjects (excluding their own) whose logarithm—logarithms are more suitable because humans perceive numbers logarithmically (order of magnitudes) [46]—are closest to the *median* log estimate $m$ of the group. This selection method thus selects central values of the distribution and removes extreme values. Since median estimates in estimation tasks are typically (but not always) closer to the true value than randomly selected estimates (Wisdom of Crowds) [47–49], we expected higher improvements in accuracy than in the Random treatment.

- *Shifted-Median* treatment: as detailed above, humans have a tendency to underestimate large quantities in estimation tasks. Recent works have suggested aggregation measures taking this bias into account, or the possibility to counteract it using artificially generated social information [20, 27]. Building on this, we here tested a method that exploits prior knowledge on this underestimation bias, by selecting estimates that are likely to reduce its effects. We define, for each group and each question, a shifted (overestimated) value $m'$ of the median log estimate $m$ that approximates the log of the true value $T$ (thus compensating the underestimation bias), exploiting a relationship between $m$ and $\log(T)$ identified from prior studies using similar tasks (for details see Experimental Design). Individuals received the estimates whose logarithm were closest to $m' > m$ (excluding their own estimate). This selection method also tends to eliminate extreme values, but additionally favors estimates that are slightly above the center of the distribution. Given the overall underestimation bias, values slightly above the center of the distribution are, on average, closer to the true value than

values at the center of the distribution. Therefore, we expected the highest improvements in collective and individual accuracy in this treatment. Note that our method uses prior domain knowledge (to estimate the true value of a quantity) but does not require a priori knowledge of the true value of the quantity at hand. That is, the accuracy of the selected estimates is a priori unknown, and they are only statistically expected to be closer to the truth.

We first describe the distributions of estimates and sensitivities to social influence in all conditions. Next, we shed light on the key effects influencing subjects' response to social information, which are: (i) the dispersion of the social information, (ii) the distance between the personal estimate and the social information, and (iii) whether the social information is higher or lower than the personal estimate. We then build a model of social information integration incorporating these findings, and use it to further analyze the impact of the number of shared estimates on social influenceability and estimation accuracy. We find, in accordance with our prediction, that improvements in collective accuracy are indeed highest in the Shifted-Median treatment, demonstrating the success of our method in counteracting the underestimation bias.

## Experimental design

Participants were 216 students, distributed over 18 groups of 12 individuals. Each individual was confronted with 36 estimation questions displayed on a tactile tablet (all questions and participants' answers are included as supplementary material). Questions were a mix of general knowledge and numerosity, and involved moderately large to very large quantities. Each question was asked twice: first, subjects were asked to provide their personal estimate $E_\text{p}$. Next, they received as social information the estimate(s) of one or several group member(s), and were asked to provide a second estimate $E_\text{s}$ (see illustration in Fig A in S1 Appendix). When providing the social information, we varied (i) the number of estimates shown ($\tau$ = 1, 3, 5, 7, 9, or 11) and (ii) how they were selected (Random, Median, or Shifted-Median treatments). The subjects were not aware of the three different treatments and were simply told that they would receive $\tau$ estimates from the other participants. Each group of 12 individuals experienced each of the 18 unique conditions (i.e., combination of number of estimates shared and their selection method) twice. Across all 18 groups, each of the 36 unique questions was asked once at every unique condition, resulting in $12 \times 36 = 432$ estimates per condition (both before and after social information sharing). Students received course credits for participation and were, additionally, incentivized based on their performance. Full experimental details can be found in S1 Appendix.

Note that a similar experimental design, using similar questions, was used in three previous studies—by partly the same authors [20–22]. We will regularly refer to and make comparisons with these studies, in particular when describing the model, as the model shares the same backbone structure as in the three other studies.

## Compensating the underestimation bias

Previous research on estimation tasks has shown that the distributions of raw estimates is generally right skewed, while the distribution of their logarithm is much more symmetric [19, 31, 39, 50]. Indeed, when considering large values, humans tend to think in terms of order of magnitude [46], making the logarithm of estimates a natural quantity to consider in estimation tasks [20]. Because the distributions of log-estimates are usually close to symmetric, the distance between the center of these distributions and the truth is often used to measure the quality of collective judgments in such tasks (Wisdom of Crowds) [21]. Although the mean is

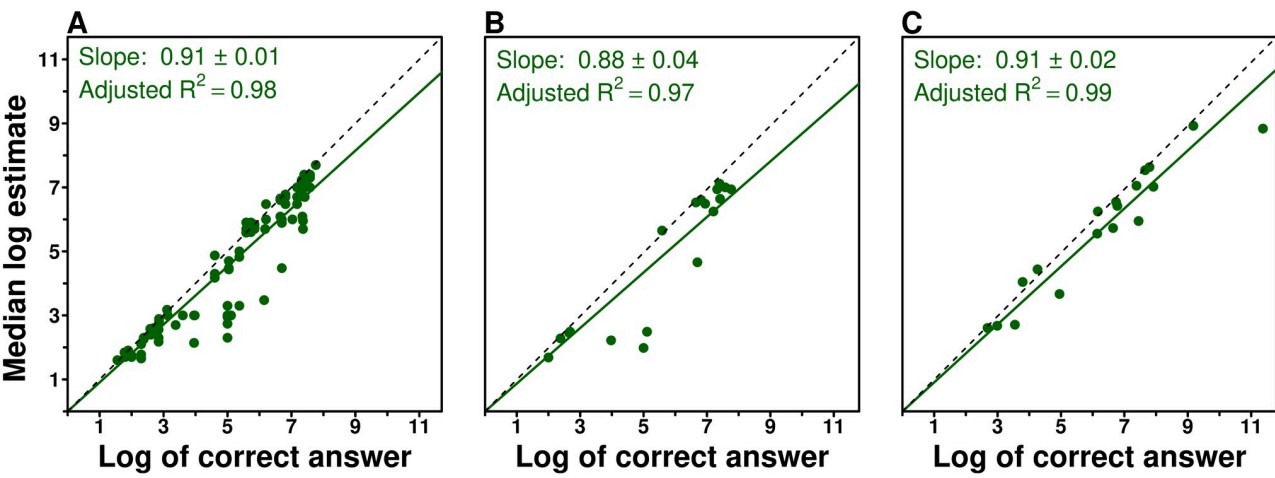

**Fig 1. The relationship between the logarithm of the correct answer and the median of the logarithm of estimates for (a) 98 questions (one dot per question) taken from a former study [20] and (b, c) 36 questions from the current experiment.** Among the 36 questions, 18 were already asked in the above cited study (b) and 18 were new (c). The slopes of the linear regression lines are 0.91 (a), 0.88 (b) and 0.91 (c), underlining the robustness of this linear trend. Note that slopes lower than 1 reflect the underestimation bias.

sometimes used to estimate the center of the distributions of log-estimates, the median is generally a better estimate of it [51], as most distributions are closer to Laplace distributions than to Gaussian distributions [52] (the median and the mean are the maximum likelihood estimators of the center of Laplace and Gaussian distributions, respectively).

Fig 1A shows that, within our domain (data taken from a previous study [20]), there is a linear correlation between the median log estimate $m$ and the log of the true value $T$: $m \sim \gamma \log(T)$, where $\gamma \approx 0.9$ is the slope expressing this correlation (the "shifted-median parameter"). In particular, the underestimation bias translates into a value of $\gamma < 1$. We found a similar linear relationship in our current study, both when using the same questions as used previously [20] (half of our questions; Fig 1B), and when using new questions (other half; Fig 1C), underlining its consistency. Fig B in S1 Appendix shows that this correlation is also found for general knowledge and numerosity questions, as well as for moderately large and very large quantities.

In the following, we quantify the significance of a statement by estimating, from the bootstrapped distribution of the considered quantity, the probability $p_0$ that the opposite statement is true (see Materials and methods for more details). By analogy with the classical $p$-value, we chose to define the significance threshold $p_0 < 0.05$. By obtaining the probability $p_0$ that the slopes in Fig 1 are larger than 1, we thus find that slopes are significantly lower than 1 ($p_0 = 0$ for 100,000 bootstrap runs in all three cases, see Fig C top row in S1 Appendix). Likewise, by calculating the probability $p_0$ that the regression slope in one panel in Fig 1 is lower than in another panel (i.e., that their difference is negative), we find that slopes are not significantly different from one another (difference between panel a and panel b: $p_0 = 0.2$; difference between panel c and panel a: $p_0 = 0.37$; difference between panel c and panel b: $p_0 = 0.17$; see Fig C bottom row in S1 Appendix).

For each group and each question, we used this linear relationship to construct a value $m'$ (the "shifted-median value") aimed at compensating the underestimation bias, i.e., to approximate the (log of the) truth: $m' = m/\gamma \sim \log(T)$, with $\gamma = 0.9$. $m'$ then served as a reference to select the estimates provided to the subjects in the Shifted-Median treatment.

## Results

### Distribution of estimates

Following previous studies where participants had to estimate a similarly various set of quantities, we use the quantity $X = \log\left(\frac{E}{T}\right)$ to represent estimates, where $E$ is the actual estimate of a quantity and $T$ the corresponding true value [20–22]. This normalization ensures that estimates of different quantities are comparable, and represents a deviation from the truth in terms of orders of magnitude. In the following, we will, for simplicity, refer to $X$ as "estimates", with $X_p$ referring to personal estimates and $X_s$ to second estimates. Fig 2 shows the distributions of $X_p$ (filled dots) and $X_s$ (empty dots) in each treatment and number of shared estimates $\tau$.

Confirming previous findings, we find narrower distributions after social information sharing across all 18 conditions (see Fig D in S1 Appendix). This narrowing amounts to second estimates $X_s$ being, on average, closer to the truth than the $X_p$. The model distributions of $X_p$ (solid lines in Fig 2) are simulated by drawing the $X_p$ from Laplace distributions, the center (median) and width (average absolute deviation from the median) of which are taken from the experimental distribution of estimates for each question. The model distributions of $X_s$ (dashed lines) are the predictions of our model presented below. One additional constraint was added in our simulations of both personal and second estimates: since in our experiment, actual estimates $E_{p,s}$ are always greater than 1, we imposed that $X_{p,s} > -\log(T)$, leading to a faster decay of the distribution for large negative log estimates. Previous studies have shown that distributions of estimates are indeed well approximated by Laplace distributions [21, 52], and [21] presented a heuristic argument to explain the occurrence of such Laplace distributions in the estimation task context. In Fig E in S1 Appendix, we show the distribution of $X_p$

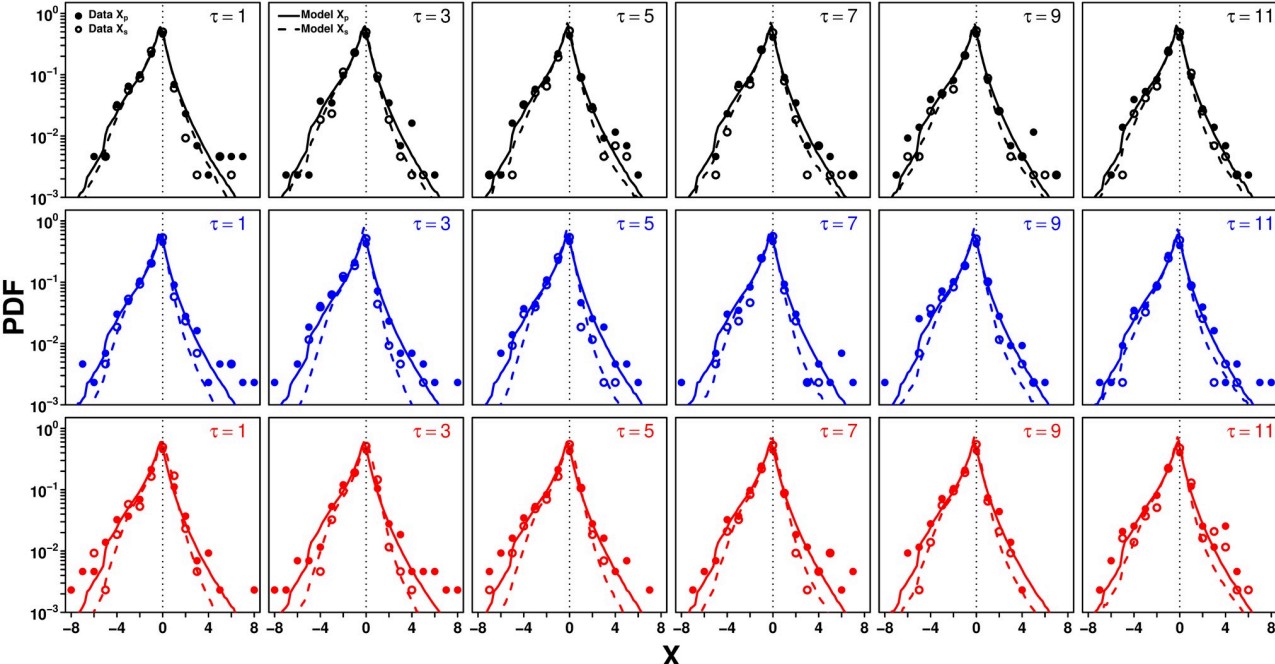

**Fig 2. Probability density function (PDF) of personal estimates $X_p$ (filled dots and solid lines) and second estimates $X_s$ (empty dots and dashed lines) in the Random (black), Median (blue), and Shifted-Median (red) treatments, for each value of $\tau$.** Dots are the data and lines correspond to model simulations.

when all conditions are combined. The good agreement between the data and the simulation further supports the Laplace distributions assumption.

## Distribution of sensitivities to social influence $S$

Consistent with heuristic strategies under time and cognitive constraints [53–55], we assume that subjects, in evaluating a series of estimates, focus on the *central tendency* and *dispersion* of the estimates that they receive as social information. These assumptions are also supported by other studies on estimation tasks [40, 56, 57]. Consistent with the logarithmic representation and Laplace distribution assumptions, we quantify the perceived central tendency and dispersion by the mean and average absolute deviation from the mean of the logarithms of the pieces of social information received, respectively.

We interpret a subject's second estimate $X_s$ as the weighted arithmetic mean (the arithmetic mean of the logs is equivalent to the log of the geometric mean) of their personal estimate $X_p$ and of the mean $M = \log(G)$ of the estimates received ($G$ is the geometric mean of the actual estimates received): $X_s = (1 − S)X_p + SM$, where $S$ is defined as the weight subjects assign to $M$, which we will call the *sensitivity to social influence*. $S$ can thus equivalently be expressed as $S = (X_s − X_p)/(M − X_p)$. $S = 0$ implies that a subject keeps their personal estimate, and $S = 1$ that their second estimate equals the geometric mean of the estimates received. As we will show below, $S$ depends on the number of estimates received and their dispersion.

In the following analysis of $S$, we will restrict $S$ to the interval [-1, 2] (for plotting reasons, we actually restrict $S$ to the interval [-1.05, 2.05]), thereby removing large values of $S$ that may disproportionately affect measures based on $S$, in particular its average. Such large values of $S$ are indeed meaningless as they are contingent on the way $S$ is defined, and do not reflect a massive adjustment from $X_p$ to $X_s$. Consider, for example, the case where $X_p = 5$ and $M = 5.001$. Then, $X_s = 5.1$ gives $S = 100$, while $X_s$ is not very different from $X_p$. Such a restriction amounts to removing about 5.3% of the data.

Fig 3 shows that the distribution of $S$, in all treatments and values of $\tau$, consists of a peak at $S = 0$ and a part that resembles a Gaussian distribution. Fig F in S1 Appendix shows the distributions of the fraction of cases where $S = 0$ per participant and per question, along with the model predictions (see the section devoted to the model). The distribution for participants is broad, but the fair agreement between the model and the experimental data suggests that this variability could mainly result from the probabilistic nature of the distribution of $S$, and not necessarily from a possible (and likely) variability of the participants' individual probability to keep their personal estimate (denoted $P_0$ below). On the other hand, the variability of the fraction of cases where $S = 0$ is much lower between questions than between participants, in both experiment and model, although the agreement there is only qualitative.

We thus assume that with a constant probability $P_0$, subjects keep their initial estimate ($S = 0$), and with probability $P_g$, they draw an $S$ in a Gaussian distribution of mean $m_g$ and standard deviation $\sigma_g$. This assumption imposes the following relation:

$$\langle S \rangle = P_g\, m_g, \quad \text{i.e.,} \quad P_g = \langle S \rangle / m_g. \tag{1}$$

To determine the values of $P_g$, $m_g$ and $\sigma_g$ per condition (i.e., treatment and value of $\tau$), we fit the distributions of $S$ with the following distribution (using the "nls" function in R):

$$f(S) = (1 − P_g)\, \delta(S) + P_g\, \Gamma(S, m_g, \sigma_g), \text{with} \tag{2}$$

$$\Gamma(S, m_g, \sigma_g) = \frac{1}{\sqrt{2\pi}\,\sigma_g} \exp\left[ -\frac{(S − m_g)^2}{2\,\sigma_g^2} \right], \tag{3}$$

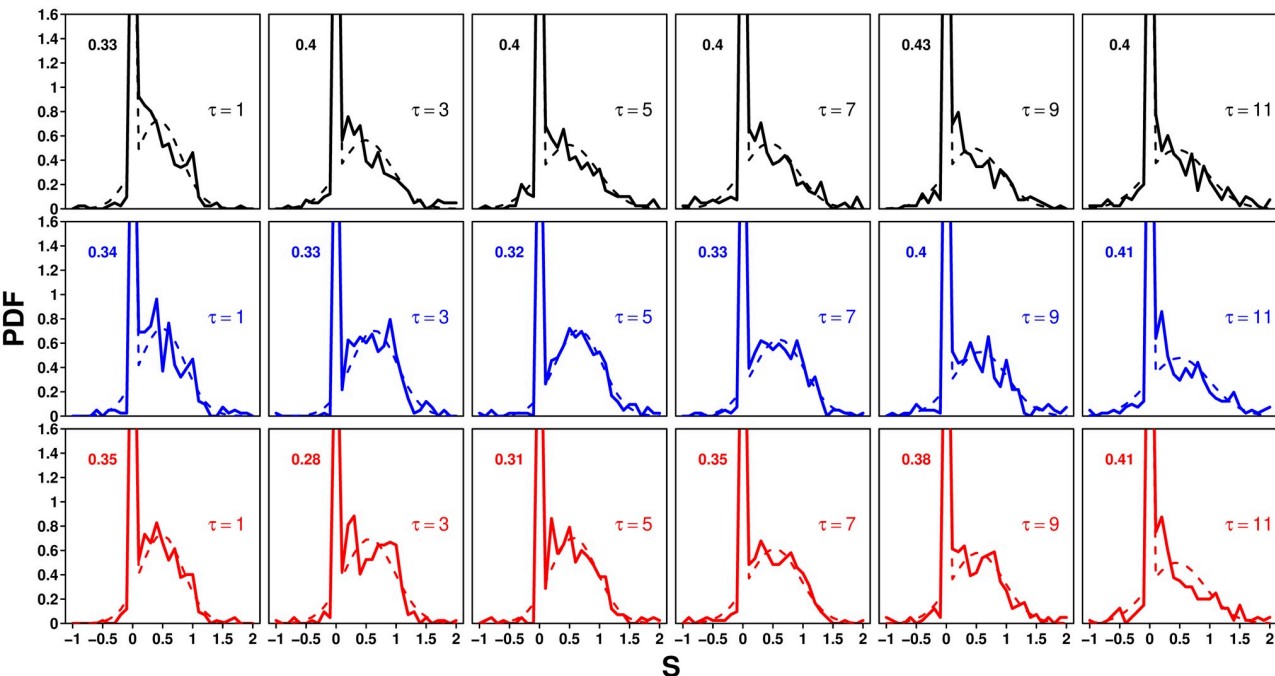

**Fig 3. Probability density function (PDF) of sensitivities to social influence S in the Random (black), Median (blue), and Shifted-Median (red) treatments, for each value of τ.** Solid lines are experimental data, and dashed lines fits using Eq 2. The experimental probability $P_0$ to keep one's personal estimate ($S = 0$) is shown in the top left corner of each graph.

where $P_g$ is fixed by Eq 1, $\delta(S)$ is the Dirac distribution centered at $S = 0$, and $\Gamma(S, m_g, \sigma_g)$ is the Gaussian distribution of mean $m_g$ and standard deviation $\sigma_g$.

Note that in [20, 21], another peak was measured at $S = 1$, amounting to about 4% of answers. However, in our experiments, this peak was absent in almost all conditions, because when more than one estimate is shared, the second estimate is very unlikely to land exactly on the geometric mean of the social information. We, therefore, did not include it in the fit.

We next analyze the dependence of the fitted parameters $P_g$, $m_g$ and $\sigma_g$ on $\tau$ in the three treatments.

## Dependence of $P_g$, $m_g$ and $\sigma_g$ on $\tau$

Fig 4 shows $P_g$, $m_g$ and $\sigma_g$ against $\tau$ in each treatment. At $\tau = 1$, values are comparable in all treatments. At intermediate values of $\tau$ ($\tau = 3, 5, 7,$ or $9$), we however observe differences between treatments, especially for $P_g$ and $m_g$. Similar to above, we quantify the significance of these treatment differences at intermediate values of $\tau$ by calculating (from the bootstrapped distribution of the considered quantity) the probability $p_0$ that the sign of the difference in the average value of the quantity at hand (here $P_g$, $m_g$ or $\sigma_g$) is opposite to that of our claim. For instance, if our claim is that $P_g$ is higher in the Shifted-Median treatment than in the Random treatment (namely, the average value of $P_g$ in the Shifted-Median treatment minus that in the Random treatment is positive), then $p_0$ is the probability that the average value of $P_g$ in the Shifted-Median treatment minus that in the Random treatment is negative (see below). Note that we compare quantities at the treatment level. Namely, we compare *functions* of $\tau$, and not *individual* values of $\tau$ (find more details in the Materials and methods).

We find that $P_g$ and $m_g$ are significantly higher in the Median ($P_g : p_0 = 0.0016$; $m_g : p_0 = 0$) and Shifted-Median ($P_g : p_0 = 0$; $m_g : p_0 = 0.003$) treatments than in the Random treatment,

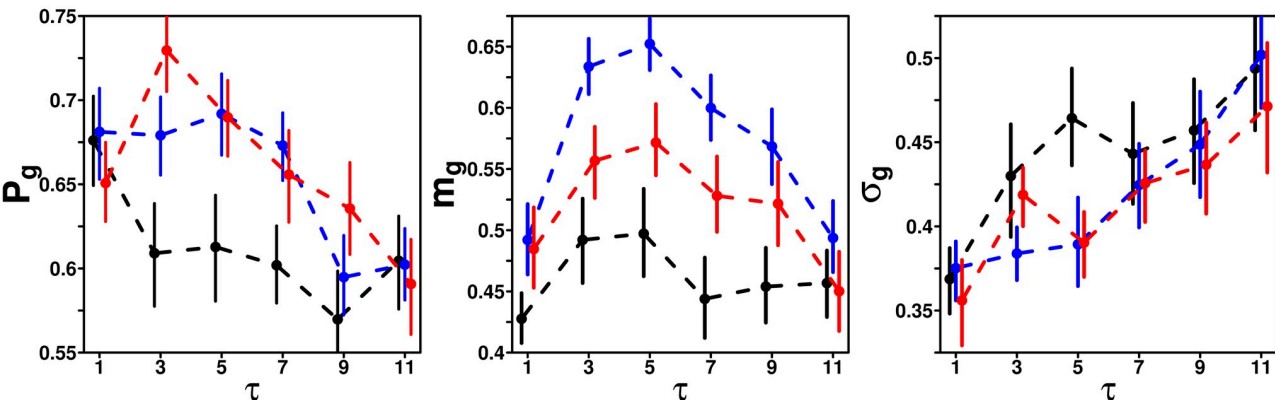

**Fig 4. $P_g$, $m_g$ and $\sigma_g$ against the number of shared estimates $\tau$, in the Random (black), Median (blue), and Shifted-Median (red) treatments.** Error bars are computed using a bootstrap procedure described in the Materials and methods, and roughly represent one standard error.

indicating a higher tendency to follow the social information in these treatments. We also find that $\sigma_g$ is significantly lower in the Median and Shifted-Median treatments than in the Random treatment (Median: $p_0 = 0.013$; Shifted-Median: $p_0 = 0.029$). Moreover, $m_g$ is significantly higher in the Median treatment than in the Shifted-Median treatment ($p_0 = 0$). However, no significant difference in $P_g$ ($p_0 = 0.13$) and $\sigma_g$ ($p_0 = 0.34$) is found between the Median and the Shifted-Median treatments. The bootstrapped distributions underlying the calculation of $p_0$ in all cases are given in Fig G in S1 Appendix. Finally, at $\tau = 11$ the three measures are similar across treatments. This was expected since all three treatments are equivalent in this case (i.e., subjects receive all pieces of social information). Note that in [22], a similar Random treatment was conducted, leading to very similar results.

## Dependence of the dispersion of the social information $\sigma$ on $\tau$

One major difference between treatments that could help explain the above results lies in the dispersion of the estimates received as social information $\sigma = \langle |X_{SI} - M| \rangle$, where $X_{SI}$ denotes the estimates received as social information. Recall that the estimates received in the Median and Shifted-Median treatments were selected by proximity to a specific value (see Experimental Design), and are thus expected to be, on average, more similar to one another (i.e., to have a lower dispersion) than in the Random treatment. Fig 5 shows that, as expected, the average dispersion $\langle \sigma \rangle$ is substantially lower in the Median and Shifted-Median treatments than in the Random treatment.

Moreover, $\langle \sigma \rangle$ increases with $\tau$ in these treatments, while it remains close to constant in the Random treatment. Expectedly, $\langle \sigma \rangle$ reaches a similar value in all treatments at $\tau = 11$. We thus expect the dependence of $P_g$, $m_g$ and $\sigma_g$ on $\tau$ observed in Fig 4 to be mediated by a dependence of these measures on $\sigma$. Note that our model reproduces the empirical patterns of Fig 5 very well. We use a "Goodness-of-Fit" (GoF) to quantify this agreement:

$$\text{GoF} = \sqrt{\frac{1}{N_\tau} \sum_\tau \frac{(O_\tau - M_\tau)^2}{C_\tau^{\,2}}}, \tag{4}$$

where $N_\tau$ is the total number of observables (e.g., 5 values of $\tau$ here), $O_\tau$ and $M_\tau$ are respectively the measured observable and the model prediction at any given $\tau$, and $C_\tau = \frac{\sigma_\tau^+ + \sigma_\tau^-}{2}$, where $\sigma_\tau^+$ and $\sigma_\tau^-$ are the upper and lower parts of the error bars (the computation of which is detailed in the Materials and methods) at each $\tau$. This measure is analogous to the reduced $\chi$-squared

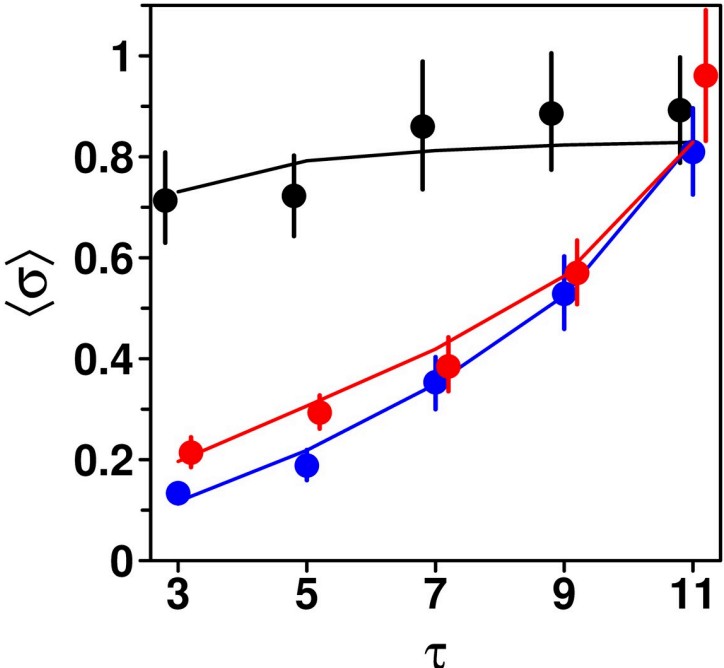

**Fig 5. Average dispersion $\langle\sigma\rangle$ of the estimates received as social information against the number of shared estimates $\tau$, in the Random (black), Median (blue), and Shifted-Median (red) treatments.** $\langle\sigma\rangle$ is mostly independent of $\tau$ in the Random treatment, while it increases with $\tau$ in the Median and Shifted-Median treatments. Dots and error bars are the data, and solid lines correspond to model simulations.

(where errors are assumed to follow Gaussian distributions), and compares the accuracy of the model predictions to the observed fluctuations in the data. Reliable model predictions should result in a GoF of order 1, which is the case for all our figures. In addition to the GoF, we provide the relative error between the model predictions and the observed data, given by:

$$\text{Relative Error} = \frac{1}{N_\tau}\sum_\tau \frac{|O_\tau - M_\tau|}{|M_\tau|}. \tag{5}$$

The GoF values and relative errors are provided in Table B in S1 Appendix.

## Dependence of $P_g$, $m_g$ and $\sigma_g$ on the dispersion $\sigma$

Fig 6 shows $P_g$, $m_g$ and $\sigma_g$ as functions of the average dispersion of estimates received as social information $\langle\sigma\rangle$, for each combination of treatment and value of $\tau$.

We find that $P_g$ and $m_g$ decrease linearly with $\langle\sigma\rangle$, reflecting a decreasing tendency to compromise with the social information as the dispersion of estimates received increases. On the contrary, $\sigma_g$ increases linearly with $\langle\sigma\rangle$, suggesting that the diversity of subjects' responses to social influence increases with the diversity of pieces of social information received. Note that in the Random treatment, the linear fits are less significant, in particular due to the fact that the range in $\langle\sigma\rangle$ is smaller than for the two other treatments. Yet, we will also implement such linear relations for the Random treatment in the model presented below, although its impact should be weaker than for the other two treatments for which $\langle\sigma\rangle$ spans a much wider range.

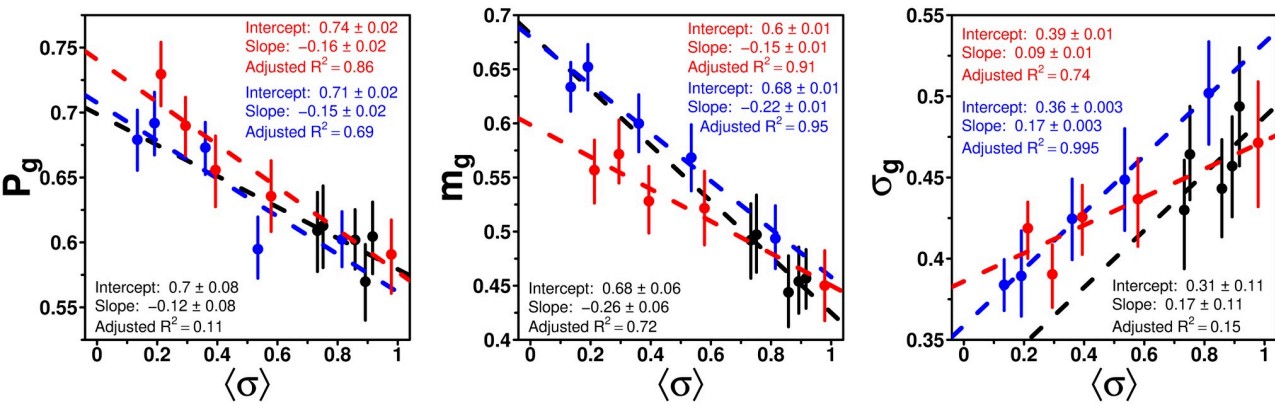

**Fig 6. $P_g$, $m_g$ and $\sigma_g$ against the average dispersion of estimates received as social information $\langle\sigma\rangle$, in the Random (black), Median (blue), and Shifted-Median (red) treatments.** Each dot corresponds to a specific value of $\tau$. Values at $\tau = 1$ were excluded since there is no dispersion at $\tau = 1$. Dashed lines show linear fits per treatment.

## Dependence of $S$ on the dispersion $\sigma$: Similarity effect

As described above, $P_g$ and $m_g$ combined determine the average sensitivity to social influence. Fig 7 shows how $\langle S\rangle = P_g\,m_g$—with the values for $P_g$ and $m_g$ taken from Fig 6A and 6B—varies with the average dispersion of estimates received $\langle\sigma\rangle$.

Consistent with $P_g$ and $m_g$ in Fig 6, $\langle S\rangle = P_g\,m_g$ decreases linearly with $\langle\sigma\rangle$ in all treatments. We call this the *similarity effect*. Moreover, this linear dependence of $\langle S\rangle$ on $\sigma$ appears to be treatment-independent, as a linear regression over all points fits the data very well.

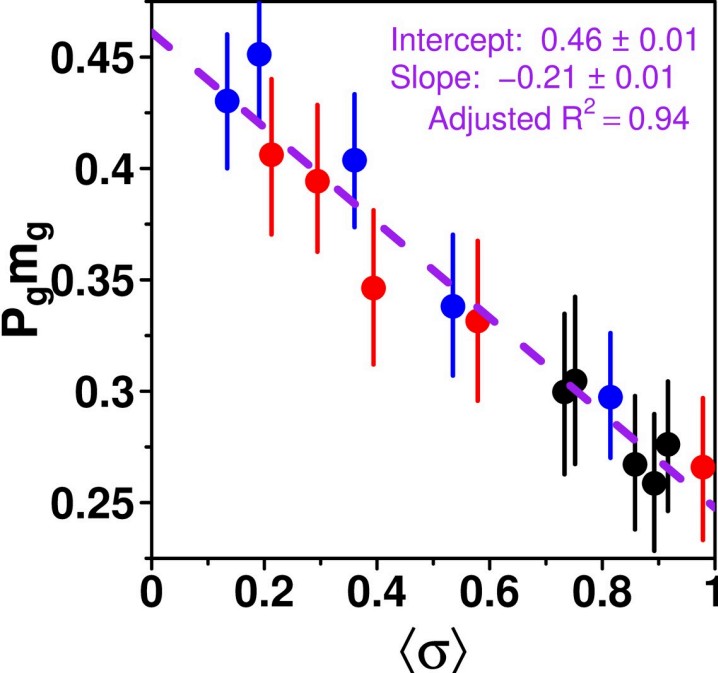

**Fig 7. $P_g\,m_g$ against the average dispersion of estimates received as social information $\langle\sigma\rangle$ in the Random (black), Median (blue), and Shifted-Median (red) treatments.** Each dot corresponds to a specific value of $\tau$. The purple dashed line shows a linear regression over all points: $P_g\,m_g$ decreases linearly with $\langle\sigma\rangle$.

　　　　　　

Note that since we found a linear dependence of $P_g$ ($P_g = a + b \langle \sigma \rangle$) and $m_g$ ($m_g = a' + b' \langle \sigma \rangle$) on $\langle \sigma \rangle$, the dependence of $\langle S \rangle = P_g m_g$ on $\langle \sigma \rangle$ could have been quadratic. Yet, the quadratic term $bb' \langle \sigma \rangle^2$ is of the order $0.2 \times 0.2 \times 0.5^2 = 0.01$, and thus negligible.

## Dependence of $S$ on $D = M - X_p$: Distance and asymmetry effect

In [20, 21], where subjects received as social information the average estimate of other group members, $S$ depended linearly on the distance $D = M - X_p$ between the personal estimate $X_p$ and the average social information $M$. This effect is known as the *distance effect*:

$$\langle S \rangle (D) = \alpha + \beta \, |D|. \qquad (6)$$

Fig 8 shows the distance effect for each condition, showing that the further the social information is away from the personal estimate, the stronger it is taken into account.

For each condition (and in agreement with [22]), we find that the center of the cusp relationship is located at $D = D_0 < 0$, rather than at $D = 0$. Moreover, the left and right slopes (coined $\beta_-$ and $\beta_+$ respectively) are not always similar, requiring us to fit the slopes separately. These combined effects result in an asymmetric use of social information, whereby social information that is higher than the personal estimate is weighted more than social information that is lower than the personal estimate. This effect is known as the *asymmetry effect*, and we will discuss it in more details below.

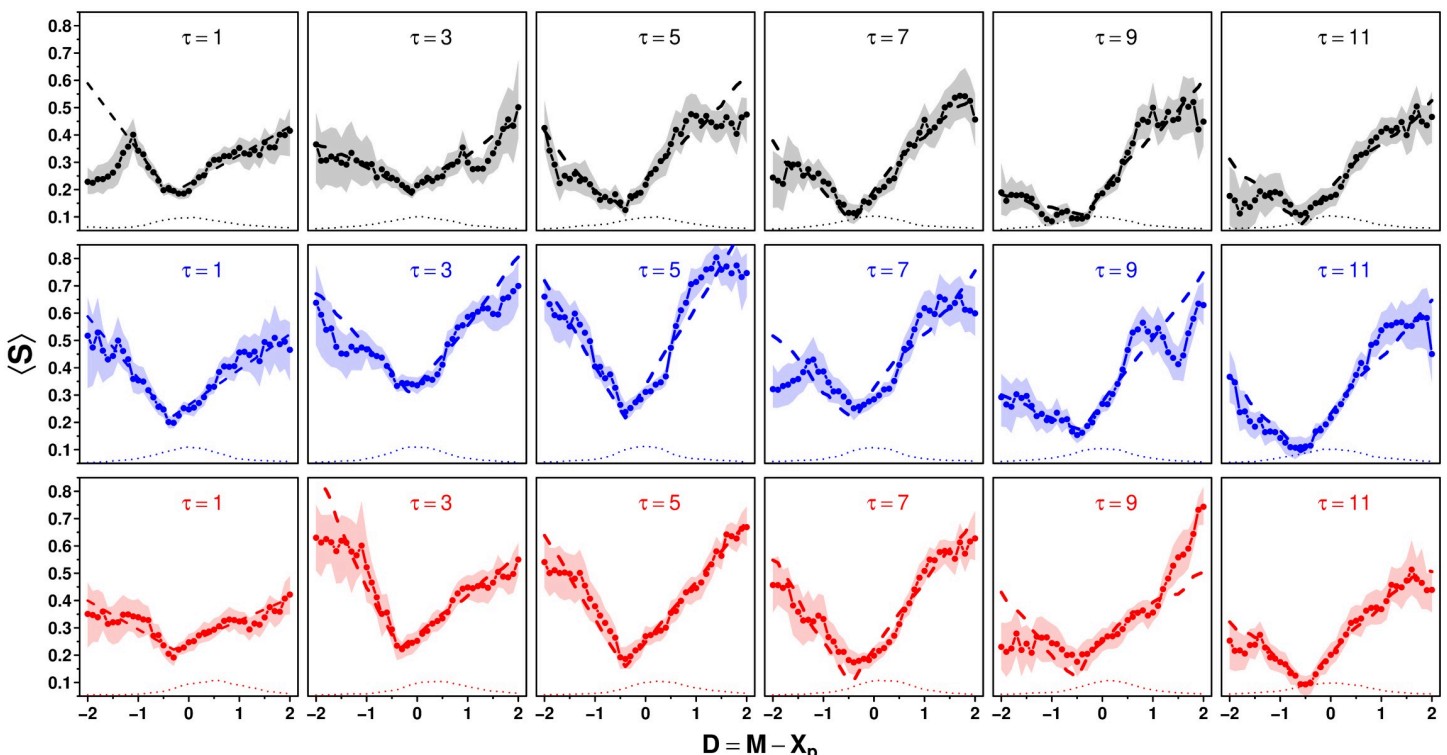

**Fig 8. Average sensitivity to social influence $\langle S \rangle$ against the distance $D = M - X_p$ between the personal estimate $X_p$ and average social information $M$, in the Random (black), Median (blue), and Shifted-Median (red) treatments for all values of $\tau$.** Dots are the data, and shaded areas represent the error (computed using a bootstrap procedure described in the Materials and methods) around the data. Dashed lines are fits using Eq 7, and dotted lines at the bottom of each panel show the density distribution of the data (in arbitrary units).

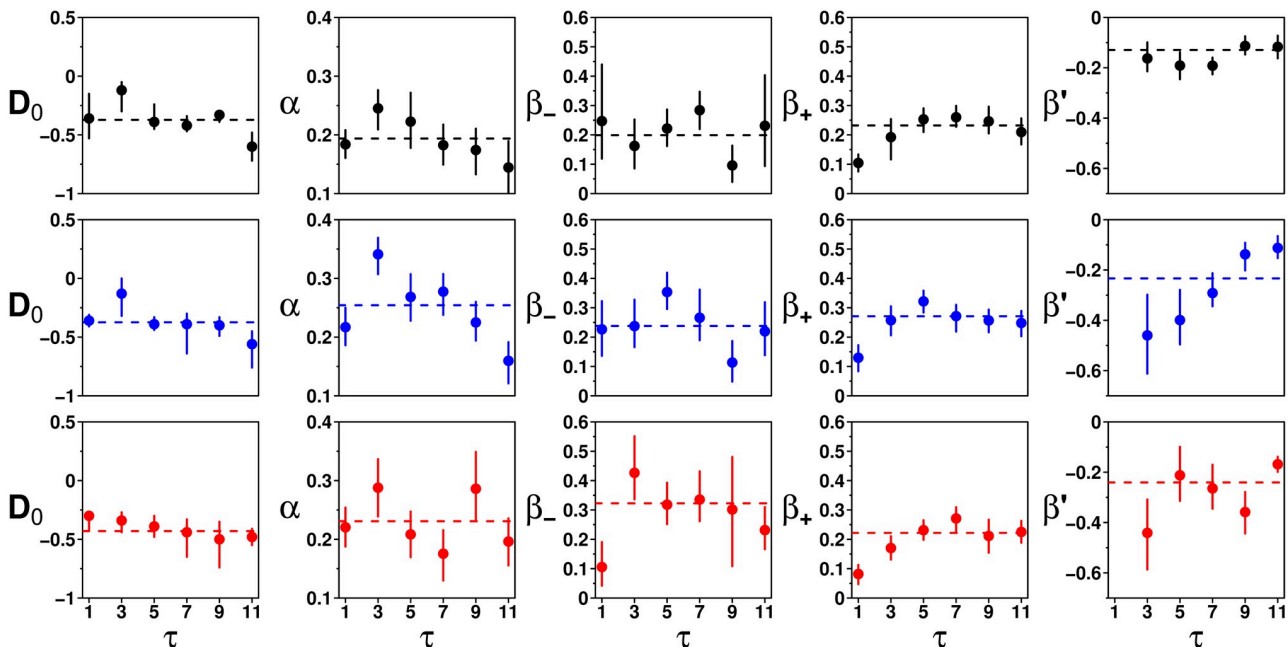

**Fig 9. Fitted parameter values of $D_0$, $\alpha$, $\beta_-$, $\beta_+$, and $\beta'$ against $\tau$ in the Random (black), Median (blue), and Shifted-Median (red) treatments.** Dashed lines correspond to the average over all values of $\tau > 1$. Parameters do not show any clear dependence on $\tau$ in each treatment (except possibly for $\beta'$ in the Median treatment) and are taken as independent of $\tau$ in the model, equal to their experimental mean.

Finally, Fig 7 showed that we need to consider the dependence of $\langle S \rangle$ on $\sigma$. Following Fig 7, we assume this dependence to be linear (with slope $\beta'$). Taking these results together, we thus arrive at the following fitting function:

$$\langle S \rangle (D, \sigma, \tau) = \alpha(\tau) + \beta_\pm(\tau) \, |D - D_0(\tau)| + \beta'(\tau) \, \sigma, \tag{7}$$

where $\alpha$, $\beta_\pm$, $\beta'$ and $D_0$ can *a priori* depend on $\tau$. At $\tau = 1$, $\sigma = 0$, therefore, $\beta'$ was excluded from the parameter fitting for this case. Further details of the fitting procedure are provided in the Materials and methods.

Fig 9 shows the fitted values against $\tau$ for each treatment, and suggests that these parameters do not systematically vary with $\tau$. We next introduce a model of social information integration, in which we will, therefore, assume that these parameters are independent of $\tau$, and equal to their average (when $\tau > 1$, see below).

## Model of social information integration

The model is based on Eq 7 and is an extension of a model developed in [22] (which itself builds on [20, 21]). The key effect we add is the dependence of subjects' sensitivity to social influence on the dispersion of estimates received as social information, since the Median and Shifted-Median treatments select relatively similar pieces of social information to share, which heavily impacts social influence (Figs 6 and 7).

The model uses log-transformed estimates $X$ as its basic variable, and each run of the model closely mimics our experimental design. For a given quantity to estimate in a given condition (i.e., treatment and number of shared estimates), 12 agents first provide their personal estimate $X_p$. Following Fig 2, these personal estimates are drawn from Laplace distributions, the center

and width of which are respectively the median $m_p$ and dispersion $\sigma_p = \langle |X_P - m_p| \rangle$ of the experimental personal estimates of the quantity.

Next, agents receive as social information $\tau$ personal estimates from other agents in the group, selected according to the selection procedure of the respective treatment (see Experimental Design). Following Fig 3, agents either keep their personal estimate ($S = 0$) with probability $P_0$, or draw an $S$ in a Gaussian distribution of mean $m_g$ and standard deviation $\sigma_g$ with probability $P_g$. According to Eq 1, $P_g = \langle S \rangle / m_g$, and $P_0 = 1 - P_g$. The calculation of $\langle S \rangle$ is based on the mean $M$ and dispersion $\sigma$ of these estimates received, and follows Eq 7. We thus obtain:

$$P_g(D, \sigma, \tau) = \langle S \rangle (D, \sigma, \tau)/m_g(\sigma) = (\alpha(\tau) + \beta_\pm(\tau) |D - D_0(\tau)| + \beta'(\tau)\,\sigma)/m_g(\sigma). \qquad (8)$$

Finally, once an $S$ is drawn for each agent, agents update their estimate according to:

$$X_s = (1 - S) X_p + S M. \qquad (9)$$

At $\tau = 1$, the values given to $P_g$, $m_g$ and $\sigma_g$ were taken from Fig 4. When sharing more than 1 estimate (i.e., $\tau > 1$), the linear dependencies of these parameters on the dispersion of the social information $\langle \sigma \rangle$, shown in Fig 6, were used. Similarly, the values of $D_0$, $\alpha$, $\beta_-$ and $\beta_+$ at $\tau = 1$ were directly taken from Fig 9, while values of $D_0$, $\alpha$, $\beta_\pm$ and $\beta'$ at $\tau > 1$ were averaged over $\tau$, and these averages were implemented in the model. This separation is done because the fitting was qualitatively different for $\tau > 1$ and $\tau = 1$, $\beta'$ being absent in the latter (no dispersion at $\tau = 1$).

In addition to this full model, we also evaluated two simpler models, leaving out either the similarity effect ($\beta' \sigma$ term) or the asymmetry effect ($D_0 < 0$ and $\beta_- \neq \beta_+$), to evaluate the importance of both effects in explaining the empirical patterns. Figs H, I, and J in S1 Appendix show the predictions when excluding the similarity effect, and Figs K, L, and M in S1 Appendix when excluding the asymmetry effect.

All model simulations results shown in the figures are averages over 10,000 runs. The full model reproduces well the distributions of estimates (Fig 2), and the dependence of $\langle \sigma \rangle$ on $\tau$ (Fig 5). We now use the model to analyze the impact of $\tau$ on sensitivity to social influence and estimation accuracy in each treatment.

## Impact of $\tau$ on sensitivity to social influence $S$

Fig 10A shows how $\langle S \rangle$ varies with $\tau$ in all treatments. We find that in the Median and Shifted-Median treatments, $\langle S \rangle$ increases sharply between $\tau = 1$ and $\tau = 3$, before decreasing steadily, consistent with the patterns of $P_g$ and $m_g$ in Fig 4 ($\langle S \rangle = P_g m_g$). In the Random treatment $\langle S \rangle$ is largely independent of $\tau$. At $\tau = 11$, all conditions (again) converge.

These patterns result from the similarity effect shown in Fig 10B: $\langle S \rangle$ decreases as the dispersion of estimates received increases, when $\tau > 1$. While in the Median and Shifted-Median treatments the different levels of $\tau$ correspond to different levels of dispersion (Fig 5), and thus different levels of $\langle S \rangle$, this effect is not present in the Random treatment. Note that consistently with the relation $\langle S \rangle = P_g m_g$, the experimental values in Fig 10B are the same as those of Fig 6.

The full model is in good agreement with the data (see GoF values in Table B in S1 Appendix). When removing the dependence on $\sigma$ from the model (and re-fitting the parameters accordingly), the inverse U-shape in the Median and Shifted-Median is attenuated, and the decrease of $\langle S \rangle$ with $\langle \sigma \rangle$ is underestimated (Fig H in S1 Appendix). This demonstrates that the similarity effect is key to explaining the patterns of sensitivity to social influence.

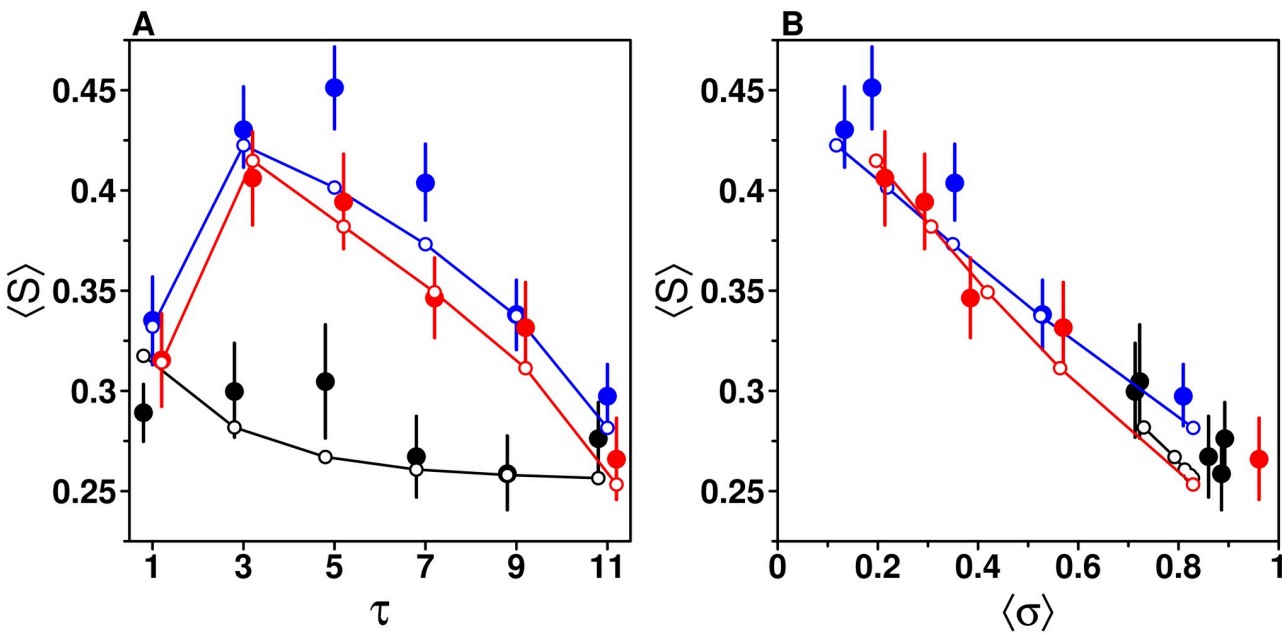

**Fig 10. Average sensitivity to social influence $\langle S \rangle$ against (a) the number of shared estimates $\tau$ and (b) the average dispersion of estimates received $\langle \sigma \rangle$, in the Random (black), Median (blue), and Shifted-Median (red) treatments.** (a) In the Random treatment, there is only a minor dependence of $\langle S \rangle$ on $\tau$. In the Median and Shifted-Median treatments, we find an inverse-U shape relationship with $\tau$. This is due to the similarity effect, as shown in (b): a linear decrease of $\langle S \rangle$ with $\langle \sigma \rangle$ when $\tau > 1$. Filled dots are the data, while empty dots and solid lines are model simulations.

### Impact of $\tau$ on $S$ when $D < 0$ and $D > 0$

A more intuitive way to understand the result that $D_0 < 0$ and $\beta_+ > \beta_-$ is that subjects' sensitivity to social influence is on average higher when $D > 0$ (i.e., when the average social information received by subjects is higher than their personal estimate) than when $D < 0$ (i.e., when the average social information received by subjects is lower than their personal estimate). Fig 11 shows this so-called *asymmetry effect*, which is fairly well captured by the full model (see GoF values in Table B in S1 Appendix).

Below, we will show that this effect also drives improvements in estimation accuracy after social information sharing. Fig L in S1 Appendix shows that the model without the asymmetry effect is unable to reproduce the higher sensitivity to social influence when $D > 0$ than when $D < 0$.

### Improvements in estimation accuracy: Herding effect

In line with [20–22], and for a given group in a given condition, we define:

- the *collective accuracy* as the absolute value of the median of all individuals' estimates of all quantities in that group and condition: $|\mathrm{Median}_{i,q}(X_{i,q})|$ (where $i$ runs over individuals and $q$ over quantities/questions);

- the *individual accuracy* as the median of the absolute values of all individuals' estimates: $\mathrm{Median}_{i,q}(|X_{i,q}|)$.

The closer to 0, the higher/better is the accuracy. Collective accuracy represents the distance of the median estimate to the truth, and individual accuracy the median distance of individual estimates to the truth (see [20] for a more detailed discussion of the interpretation of these two quantities and their differences).

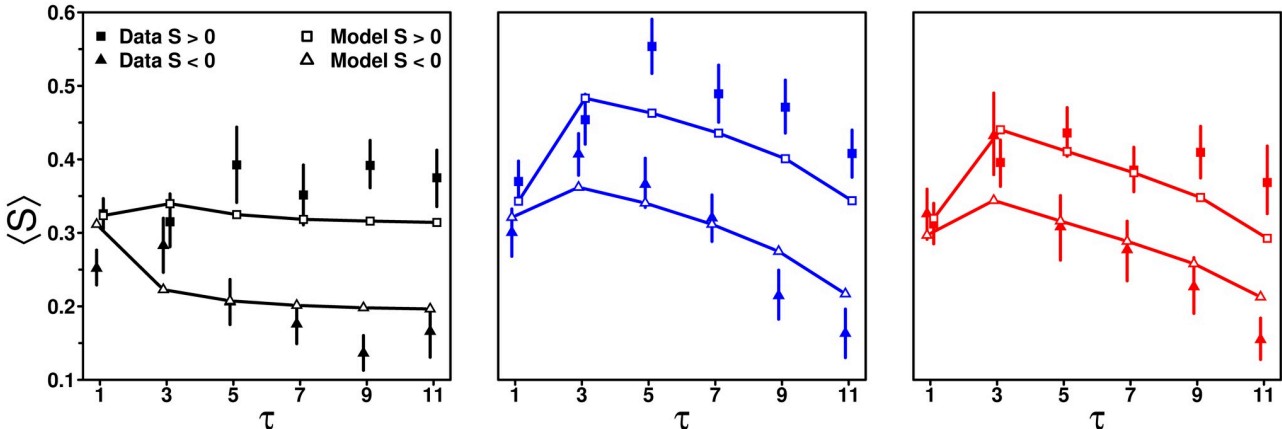

**Fig 11. Average sensitivity to social influence $\langle S \rangle$ against the number of shared estimates $\tau$, in the Random (black), Median (blue), and Shifted-Median (red) treatments, when the average social information $M$ is higher than the personal estimate $X_p$ ($D = M - X_p > 0$; squares) and when it is lower ($D < 0$; triangles).** Subjects follow the social information more on average when $M$ is higher than $X_p$, than when it is lower. Filled symbols represent the data, while solid lines and empty symbols are model simulations. Table A in S1 Appendix shows the percentage of cases when $D < 0$ and $D > 0$ in all conditions.

Fig 12 shows how collective and individual accuracy depend on $\tau$ in each treatment. Let $O_\tau$ and $O'_\tau$ denote the collective or individual accuracy for each value of $\tau$, before and after social information sharing. Since the dependence of both quantities on $\tau$ is weak compared to the size of the error bars, we here consider their average $\langle O \rangle = \frac{1}{N_\tau}\sum_\tau O_\tau$ and $\langle O' \rangle = \frac{1}{N_\tau}\sum_\tau O'_\tau$ over all values of $\tau$ ($N_\tau = 6$). We quantify the improvements in collective or individual accuracy as the positive difference between both averages: $\langle O \rangle - \langle O' \rangle$, and assess their significance by computing the probability $p_0$ that the improvement is negative.

We find that collective accuracy improves mildly—but significantly—in the Random and Median treatments ($p_0 = 0.0002$ and $0.0035$ respectively, see the bootstrapped distributions in Fig N top row in S1 Appendix), as predicted by the model.

This improvement is due to the asymmetry effect (Fig 11), which partly counteracts the human tendency to underestimate quantities [20, 27–29]. Indeed, giving more weight to social information that is higher than one's personal estimate shifts second estimates toward higher values, thus improving collective accuracy. The model without the asymmetry effect is unable to predict this improvement in collective accuracy (Fig M in S1 Appendix).

In the Shifted-Median treatment the improvement in collective accuracy is substantially higher and highly significant ($p_0 = 0$ for 10,000 bootstrap runs, see the bootstrapped distribution in Fig N top right panel in S1 Appendix). The improvement in collective accuracy is substantially (and significantly) higher in the Shifted-Median treatment than in the Random treatment ($p_0 = 0.0018$). However, we find no significant difference in the improvement between the Median and Random treatments ($p_0 = 0.47$). The corresponding bootstrapped distributions are shown in Fig O in S1 Appendix.

This higher improvement in the Shifted-Median treatment is a consequence of the selection procedure of the pieces of social information. As shown in Fig 10, participants have a tendency to partially follow the social information ($0 < \langle S \rangle < 1$ in all conditions, a.k.a. *herding effect*). Although there are no substantial differences in $\langle S \rangle$ between the Median and Shifted-Median treatments, the estimates received as social information overestimate the group median in the Shifted-Median treatment. A similar level of $\langle S \rangle$ thus shifts seconds estimates toward higher values (as compared to the Median treatment), thereby partly countering the underestimation bias and boosting collective accuracy.

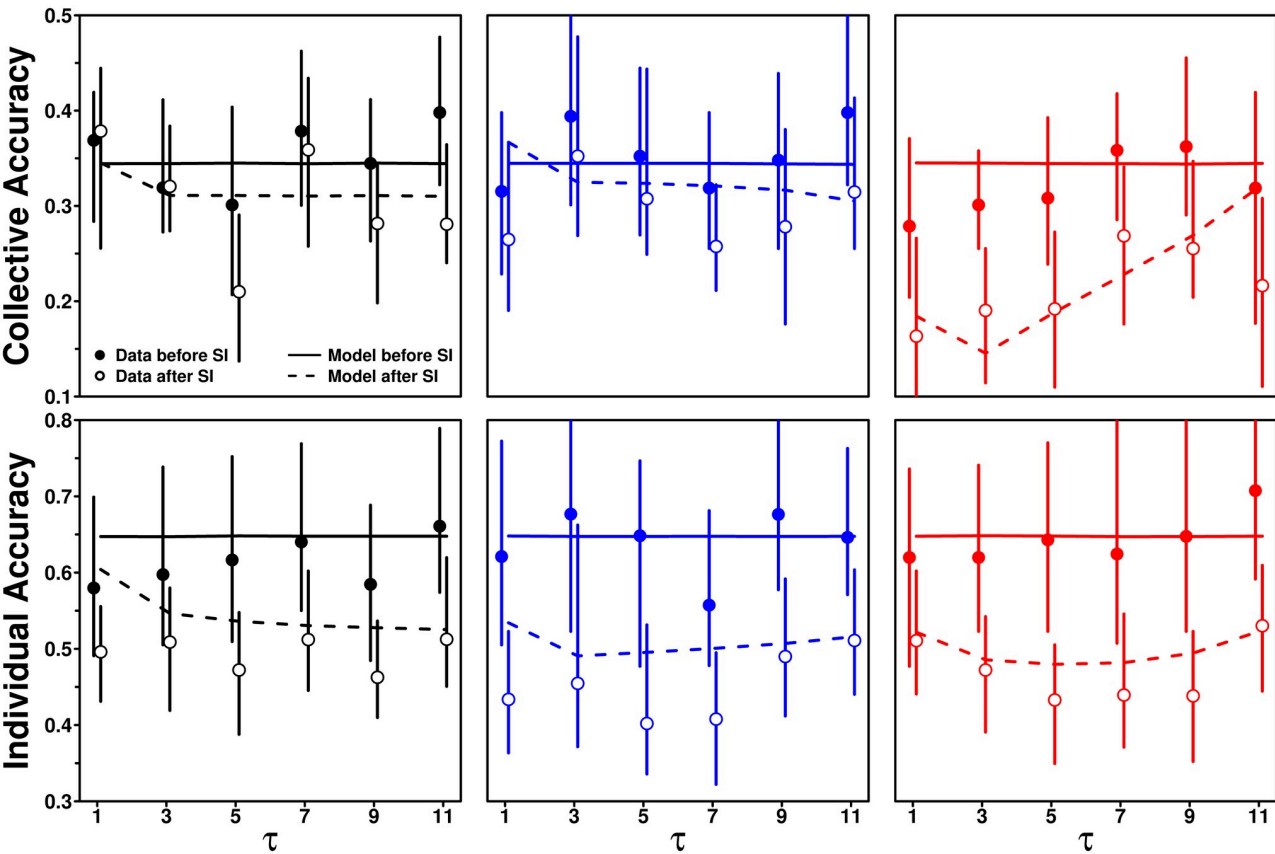

**Fig 12. Collective and individual accuracy against the number of shared estimates $\tau$, before (filled dots) and after (empty circles) social information sharing, in the Random (black), Median (blue), and Shifted-Median (red) treatments.** Values closer to 0 indicate higher accuracy. Solid and dashed lines are model simulations before and after social information sharing, respectively.

For the individual accuracy, we find substantial and significant improvements in all treatments ($p_0 = 0$, see Fig N bottom row in S1 Appendix), with slightly (and significantly) higher improvements in the Median and Shifted-Median treatments than in the Random treatment ($p_0 = 0.028$ and $0.027$ respectively, see Fig P in S1 Appendix), due to the similarity effect which boosts social information use in these treatments (Fig 10).

This confirms previous studies showing that higher levels of social information use (when $0 < \langle S \rangle < 0.5$) increase the narrowing of the distribution of estimates (Fig 2), thereby increasing individual accuracy [19, 20]. The model correctly predicts the magnitude of improvements in all treatments (see GoF values in Table B in S1 Appendix).

As a final remark, note that the size of the error bars presented in Fig 12 (and in the figures of the next sections) for each individual condition (treatment and value of $\tau$; before and after social information) could be slightly misleading. Indeed, for each bootstrap run, the 12 data points in each panel of Fig 12 are in fact highly correlated, and our paired analysis presented in Figs N and O in S1 Appendix (bootstrapped distributions of the difference between two considered observables) constitutes a more rigorous and fairer assessment of the significance of our claims presented above.

## Impact of $D$ on estimation accuracy

Because subjects behave differently when receiving social information that is higher ($D = M - X_p > 0$) or lower ($D < 0$) than their personal estimate, we next study how these different

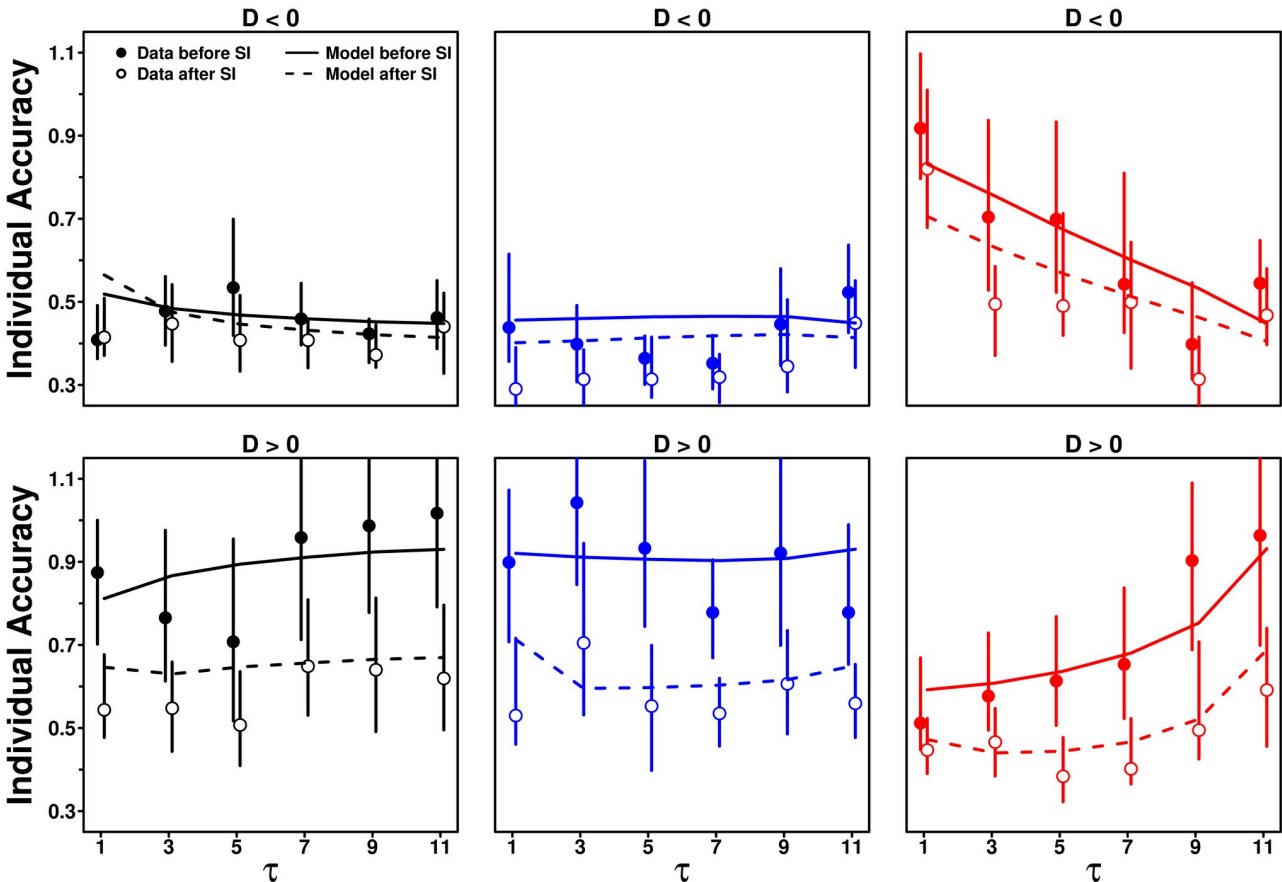

**Fig 13. Individual accuracy against the number of shared estimates $\tau$, before (filled dots) and after (empty circles) social information sharing, in the Random (black), Median (blue), and Shifted-Median (red) treatments.** The population was separated into subjects' answers where the average social information received $M$ was lower than their personal estimate $X_p$ ($D = M − X_p < 0$) and subjects' answers where the average social information received was higher than their personal estimate ($D > 0$). Solid and dashed lines are model simulations before and after social information sharing, respectively. Individual accuracy improves marginally for $D < 0$, but substantially for $D > 0$.

scenarios impact accuracy. Fig 13 shows the individual accuracy for each condition, separating the answers where the personal estimate of a subject was above or below the social information.

We find that, in the Random and Median treatments, subjects were more accurate when $D < 0$ than when $D > 0$ before social information sharing. This is a consequence of the under-estimation bias, as personal estimates in the former (latter) case are, on average, more likely to be above (below) the median estimate of the group—and therefore closer to (farther from) the truth. In the Shifted-Median treatment, however, we observe a more complex pattern: (i) at low values of $\tau$, individual accuracy is worse before social information sharing in this treatment than in the Random and Median treatments when $D < 0$, while it is better when $D > 0$. This reversed pattern suggests that the shifted-median values tend, on average, to slightly overesti-mate the truth; (ii) individual accuracy improves with $\tau$ when $D < 0$, but declines with it when $D > 0$. As $\tau$ increases, the average social information indeed decreases until it is the same as in both other treatments at $\tau = 11$. In all conditions, individual accuracy improves mildly (but sig-nificantly) after social information sharing when $D < 0$ (see Fig Q top row in S1 Appendix), while it improves substantially (and highly significantly) when $D > 0$ (see Fig Q bottom row in

S1 Appendix). The model captures the main trends (or absence of them) well (see GoF and relative errors in Table B in S1 Appendix). Fig R in S1 Appendix shows the equivalent figure for collective accuracy, showing qualitatively similar results.

Note that it may seem puzzling that accuracy before social information sharing is condition dependent. This is because we consider subpopulations, selected according to specific criteria ($D > 0$ and $D < 0$). When selecting such subpopulations, nothing forbids that inter-individual differences exist in the accuracy of personal estimates. When considering the whole population such differences between conditions, by definition, disappear (Fig 12).

## Impact of $S$ on estimation accuracy

Finally, we studied how subjects' sensitivity to social influence affects estimation accuracy, by separating subjects' answers into those for which $S$ was either below or above the median value of $S$ in that condition. Fig 14 shows individual accuracy for both categories.

Subjects in the below-median category provided more accurate personal estimates than those in the above-median category. It is well-known that more accurate individuals use less

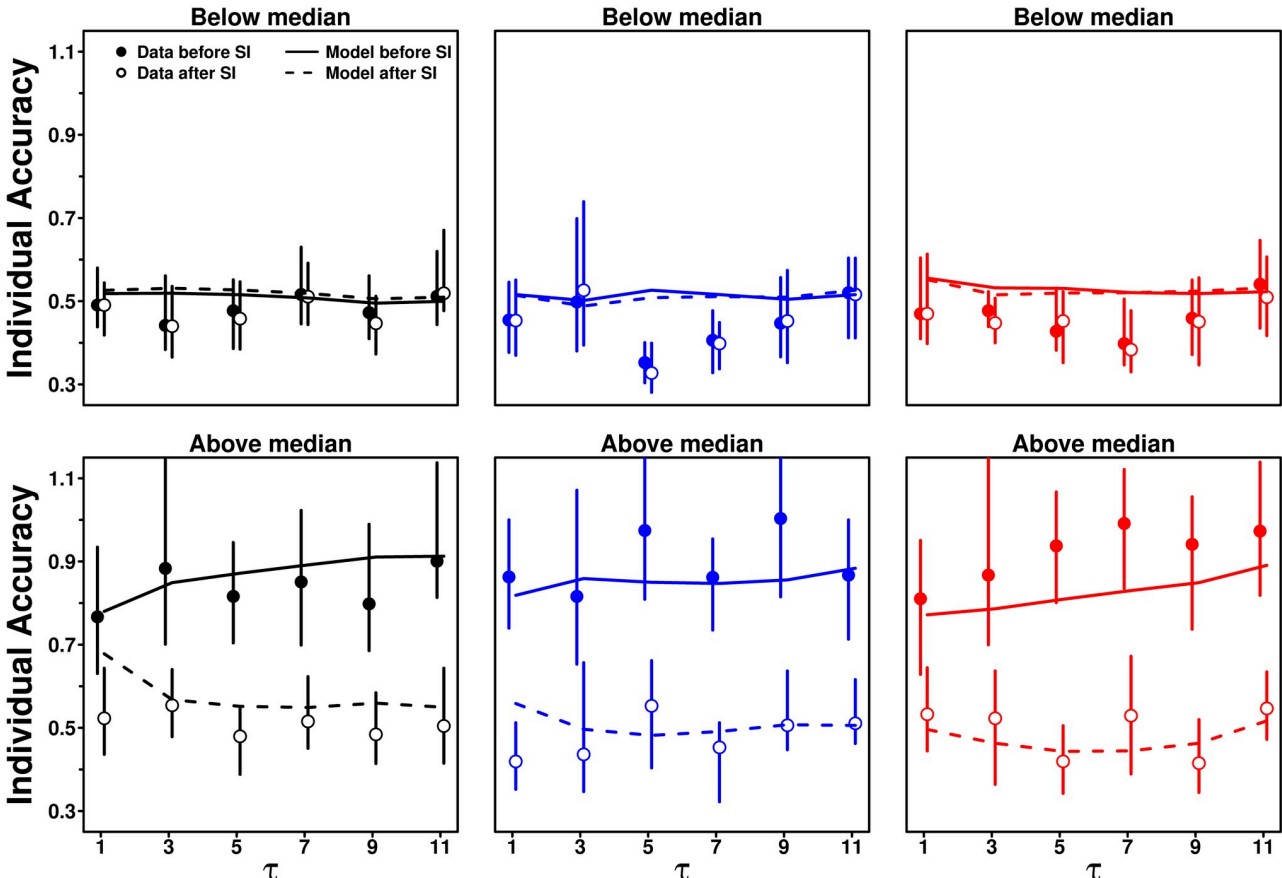

**Fig 14. Individual accuracy against the number of shared estimates $\tau$, before (filled dots) and after (empty circles) social information sharing, in the Random (black), Median (blue), and Shifted-Median (red) treatments.** In each condition, the subjects' answers were separated according to their corresponding value of $S$ with respect to the median of $S$. Solid and dashed lines are model simulations before and after social information sharing, respectively. When $S$ is lower than the median, the subjects tend to keep their initial estimate, and individual accuracy therefore does not change much. When $S$ is higher than the median, the subjects tend to compromise more with the social information, resulting in high improvements.

social information (they are also the most confident subjects in their personal estimate [20]), and this insight has also been used to improve collective estimations [37]. This result is in part related to the distance effect (Fig 8): subjects use social information the least when their initial estimate is close to the average social information, which is itself, on average, close to the truth.

Because subjects in the below-median category disregard, or barely use, social information, they do not (or barely) improve in accuracy after social information sharing. We observe no significant improvement in the Random and Median treatments ($p_0$ = 0.28 and 0.44, respectively), and a marginal improvement in the Shifted-Median treatment, although not clearly significant ($p_0$ = 0.06; see Fig S top row in S1 Appendix). On the contrary, subjects in the above-median category tend to compromise with the social information, thereby substantially improving in individual accuracy after social information sharing, and reaching similar levels of accuracy as the below-median category. Improvements are highly significant in all treatments ($p_0$ = 0, see Fig S bottom row in S1 Appendix).

The model, again, reproduces these findings well, in particular the magnitudes of improvements in all cases (see GoF and relative errors in Table B in S1 Appendix), which are also in agreement with [20–22]. Fig T in S1 Appendix shows the equivalent figure for collective accuracy, showing qualitatively similar patterns, albeit with substantially higher improvements in the Shifted-Median treatment for the above-median category, consistent with Fig 12.

## Discussion

We have studied the impact of the number of estimates presented to individuals in human groups, and of the way these estimates are selected, on collective and individual accuracy in estimating large quantities. Our results are driven by four key mechanisms underlying social information integration:

1.  subjects give more weight to the social information when the distance between the average social information and their own personal estimate increases (*distance effect*). This effect has been found in several previous studies [20–22, 27]. But note that in [17, 58], the authors found that for large distances, the opposite was true (namely, the weight given to advice decreased with distance to the personal estimate);

2.  subjects give more weight to the central tendency of multiple estimates when it is higher than their own personal estimate, than when it is lower (*asymmetry effect*). This asymmetry effect, also found in [22, 27], shifts second estimates toward higher values, thereby partly compensating the underestimation bias and improving collective accuracy. The asymmetry effect suggests that people are able to selectively use social information in order to counterbalance the underestimation bias, even without external intervention (Random treatment). Note that we cannot exclude that this effect might be partly contingent to our experimental design, and that future works find no such effect, or the opposite effect, when participants are asked to estimate different sets of quantities;

3.  subjects follow social information more when the estimates are more similar to one another (*similarity effect*). Previous studies have shown that similarity in individuals' judgments correlates with judgment accuracy [59, 60], suggesting that following pieces of social information more when they are more similar is an adaptive strategy to increase the quality of one's judgments. Our selection method in the Median and Shifted-Median treatments capitalized on this effect as it selected relatively similar pieces of social information, thereby

counteracting the human tendency to underuse social information [20, 61, 62], resulting in higher individual improvement in both treatments than in the Random treatment;

4. subjects tend to partially copy each other (*herding effect*), leading to a convergence of estimates after social information sharing, and therefore to an improvement in individual accuracy in all treatments. This effect is adaptive in most real-life contexts, as personal information is often limited and insufficient, such that relying on social information, at least partly, is an efficient strategy to make better judgments and decisions. Moreover, note that contrary to popular opinion, convergence of estimates need not yield negative outcomes (like impairing the Wisdom of Crowds [19, 32, 37]): even if the average opinion is biased, sharing opinions may temper extreme ones and improve the overall quality of judgments [63]. This tendency to follow the social information has another important consequence: it is possible to influence the outcome of collective estimation processes in a desired direction. In the Shifted-Median treatment, we showed that subjects' second estimates could be "pulled" towards the truth, thus improving collective accuracy. This is an example of *nudging*, also demonstrated in other contexts [64]. Previous studies have shown that the same tendency can also lead, under certain conditions, to dramatic situations in which everybody copies everybody else indiscriminately ("herd behavior") [65].

Next, we developed an agent-based model to study the importance of these effects in explaining the observed patterns. The model assumes that subjects have a fast and intuitive perception of the central tendency and dispersion of the estimates they receive, coherent with heuristic strategies under time and computational constraints [53–55], and consistent with previous findings [40, 56, 57]. By using simpler models excluding either the asymmetry effect or similarity effect, we demonstrated that the above effects are key to explaining the empirical patterns of sensitivity to social influence and estimation accuracy. It is conceivable that the strategies used by people when integrating up to 11 pieces of social information in their decision-making process are very diverse and complex. Yet, despite its relative simplicity, our model is able to capture all the main observed patterns, underlining the core role of these effects in integrating several estimates of large quantities.

Our goal was to test a method to improve the quality of individual and collective judgments in social contexts. The method exploits available knowledge about cognitive biases in a given domain (here the underestimation of large quantities in estimation tasks) to select and provide individuals with relevant pieces of social information to reduce the negative effects of these biases. In [21], the social information presented to the subjects was manipulated in order to improve the accuracy of their second estimates. However, at variance with our present study, the correct answer to each question needed to be known *a priori*, and was exploited by "virtual influencers" providing (purposefully) incorrect social information to the subjects, specifically selected to counter the underestimation bias. Even though such fake information can help the group perform better, our method avoids such deception, and extends to situations in which the estimation context is known, but not the truth itself. Note that our shifted-median value $\gamma$ $\approx 0.9$ aimed at approximating the truth. The results of [21] suggest that a slightly lower value of $\gamma$ (thus aiming at slightly overestimating the truth) could boost improvements in accuracy even further.

Another previous study exploited the underestimation bias by *recalibrating* personal estimates, thereby also successfully counteracting the underestimation bias [27]. Fig U in S1 Appendix compares our Shifted-Median treatment to a direct recalibration of personal estimates, where all $X_p$ are divided by $\gamma = 0.9$. Collective accuracy improves similarly under both methods. Individual accuracy, however, degrades with the recalibration method, while it strongly improves with the Shifted-Median method. Our method thus outperforms a mere

recalibration of personal estimates. Moreover, note that recalibrating initial estimates may be useful from an external assessor's point of view, but does not provide participants with an opportunity to improve their accuracy, individually or collectively.

Our method may, in principle, be applied to different domains. Future work could, for instance, test this method in domains where overestimation dominates, by defining a shifted-median value below the group median; or in domains where the quantities to estimate are negative (or at least not strictly positive) or lower than one (i.e., negative in log). Another interesting direction for future research would be to explore ways to refine our method. Figs V and W in S1 Appendix show that collective and individual accuracy improve more for very large quantities than for moderately large ones, although the levels of underestimation are similar in both cases (Fig B in S1 Appendix). This suggests that the linear relationship between the median (log) estimates and the (log of the) true value may be insufficient to fully characterize this domain of estimation tasks. Considering other distributional properties, such as the dispersion, skewness and kurtosis of the estimates received, could help to fine tune the selection method to further boost accuracy.

Finally, let us point out that our population sample consisted of German undergraduate students. In [20], a cross-cultural study was conducted in France and Japan, using a similar paradigm, and found similar levels of underestimation in both countries, albeit slightly higher levels of social information use in Japan. This suggests that our observed underestimation bias is widespread in this domain, although a systematic comparison of the levels of bias and social information use in different (sub)populations is still lacking. Filling this gap could represent a major step forward in research on social influenceability and cognitive biases.

To conclude, we believe that the mechanisms underlying social information use in estimation tasks share important commonalities with related fields (e.g., opinion dynamics [66]), and that our method has the potential to inspire research in such fields. For instance, one could imagine reducing the in-group bias by extending the amount of discrepant/opposite views presented to individuals in well-identified opinion groups. Implementing methods similar to ours in recommender systems and page-ranking algorithms may thus work against filter bubbles and echo chambers, and eventually reduce polarization of opinions [67]. Similarly, it is conceivable that the effects of well-known cognitive biases such as the confirmation [68] or overconfidence bias [69] could be dampened by strategically sharing social information.

## Materials and methods

### Computation of the error bars

The error bars indicate the variability of our results depending on the $N_Q = 36$ questions presented to the subjects. We call $x_0$ the actual measurement of a quantity appearing in the figures by considering all $N_Q$ questions. We then generate the results of $N_{exp} = 1,000$ new effective experiments. For each effective experiment indexed by $n = 1, \ldots, N_{exp}$ (bootstrap runs), we randomly draw $N'_Q = N_Q$ questions among the $N_Q$ questions asked (so that some questions can appear several times, and others may not appear) and recompute the quantity of interest which now takes the value $x_n$. The upper error bar $b_+$ for $x_0$ is defined so that $C = 68.3\%$ (by analogy with the usual standard deviation for a normal distribution) of the $x_n$ greater than $x_0$ are between $x_0$ and $x_0 + b_+$. Similarly, the lower error bar $b_-$ is defined so that $C = 68.3\%$ of the $x_n$ lower than $x_0$ are between $x_0 - b_-$ and $x_0$. The introduction of these upper and lower confidence intervals is adapted to the case when the distribution of the $x_n$ is unknown and potentially not symmetric.

## Quantification of significance

For any claim of which we want to assess the significance, we use the same bootstrap procedure as above and generate the distribution of the relevant quantity. In particular, this method allows us to study paired statistics for the difference between two relevant observables. For instance, in Fig 4, we want to check whether $P_g$ is significantly higher in the Shifted-Median treatment than in the Random treatment at intermediate values of $\tau$ ($\tau = 3$, 5, 7, 9). The quantity of interest is therefore the difference between the average value of $P_g$ (over $\tau = 3, 5, 7, 9$) in the Shifted-Median treatment and that in the Random treatment, and we want to show that this quantity is significantly positive. We generate $N_{exp} = 10,000$ bootstrap runs for the quantity of interest ($N_{exp} = 100,000$ for the distributions related to Fig 1) and obtain the distribution of its possible values. From this distribution, we can then calculate the probability $p_0$ that the difference is negative. More generally, given any claim, we check its significance by estimating the probability $p_0$ that its opposite is true. Given the similarity between this quantifier $p_0$ and the classical $p$-value (although our approach does not assume Gaussian-distributed errors), we consider a result significant whenever $p_0 < 0.05$. Note that in the context of our bootstrap approach, $p_0 = 0$ means that no occurrence of the event was observed during the $N_{exp}$ runs. Hence, the actual $p_0$ is estimated to be of the order of or lower than $1/N_{exp}$.

It is important to keep in mind that we compare functions of $\tau$, and not just results for individual values of $\tau$. Indeed, while differences for each value of $\tau$ may not always be significant, differences at the treatment level are often highly significant. Consider for instance collective accuracy in the Shifted-Median treatment of Fig 12. While error bars overlap at all levels of $\tau$, casting doubt on the improvement's significance at any individual value of $\tau$, the fact that *all* points after social information sharing are below the points before social information sharing intuitively suggests that the improvement has to be significant, as confirmed by our paired statistical analysis. In fact, even if two observables that we wish to compare fluctuate wildly between bootstrap runs, leading to significant error bars which may overlap for the two quantities (like in Fig 12), these fluctuations are in fact highly correlated and the statistics of the difference between the two quantities should then serve at quantifying their relative magnitude.

## Fitting procedure used in Fig 8

Each combination of treatment and number of shared estimates contains 432 estimates. When binning data, one has to trade off the number of bins (thus displaying more detailed patterns) and the size of the bins (thus avoiding too much noise). In Fig 8, the noise within each condition was relatively high when using a bin size below 1. However, bins of size 1 were hiding the details of the relationship between $\langle S \rangle$ and $D$, especially the location of the bottom of the cusp.

To circumvent this problem, we use a procedure that is well adapted to such situations (previously described in [22]). First, remark that a specific binning leaves one free to choose on which values the bins are centered. For instance, a set of 5 bins centered on -2, -1, 0, 1 and 2 is as valid as a set of 5 bins centered on -2.5, -1.5, -0.5, 0.5, and 1.5, as the *same* data are used in both cases. Both sets of points produced are replicates of the same data, but we now have 10 points instead of 5.

In each panel of Fig 8, we used such a moving center starting the first bin at -2, and the last one at +2, producing histograms (of bin size 1) in steps of 0.1 for the bin center. This replicated the data 9 times, thus having overall 10 replicates and 50 points, instead of 5. We then removed the values beyond $D = 2$, thus keeping 41 points ($D = -2$ to $D = 2$).

Next, we used the following functions to fit the data in Fig 8 and obtain the values of all parameters in each condition. At $\tau = 1$:

$$S_{\text{fit}} = \alpha + \beta_{\pm} |D - D_0|.$$

At $\tau > 1$ (i.e., including the dispersion $\sigma$):

$$S_{\text{fit}} = \alpha + \beta_{\pm} |D - D_0| + \beta' \, \sigma.$$

$D_0$ was first fitted separately, as the minimum of an absolute value function fitted locally with the data points shown in Fig 8, in the interval [-1.2, 0.2] in each condition. Only in the Random treatment at $\tau = 3$ and 5, and in the Median treatment at $\tau = 3$, was the upper bound taken as 0.6 instead of 0.2 (the lower bound always remained the same, i.e. -1.2).

For the fitting of $\alpha$, $\beta_{-}$, $\beta_{+}$ and $\beta'$, we used all the data comprised within the interval shown in Fig 8, namely [-2.5, 2.5] (bins are of size one, so the dot at $D = 2$, for instance, shows the average of $S$ between 1.5 and 2.5). In a few cases only did we slightly restrict the fitting interval in order to obtain better results:

- Random treatment, $\tau = 1$ and 11: [-1.65, 2.5]

- Median treatment, $\tau = 7$: [-1.9, 2.5]

- Shifted-Median treatment, $\tau = 3$: [-2, 2.5]

- Shifted-Median treatment, $\tau = 9$: [-1.2, 1.5]

We wrote a program to perform the minimization of least squares. Let $Q = \sum_i (S_i - S_{i\text{fit}})^2 = \sum_i (S_i - \alpha - \beta_{\pm} |D_i - D_0| - \beta' \, \sigma_i)^2$ be the sum ($\beta' = 0$ when $\tau = 1$), over all the data in the chosen interval (indexed by $i$), of squared distances between $S$ and $S_{\text{fit}}$. Note that the data indexed by $i$ correspond to individual participant's answers, not to the averaged values shown in Fig 8. This is why the squared distances are not weighted (all individual answers have the same weight). We then equated to 0 the partial derivatives of $Q$ with respect to $\alpha$, $\beta_{-}$, $\beta_{+}$ and $\beta'$ (when $\tau > 1$) to obtain the values of these parameters.

## Supporting information

**S1 Appendix. Details of the experimental design and supplementary figures and tables.** Further details about the experimental design are provided (including the questions asked), as well as figures and tables supporting the statistical analysis and the main discussion. **Fig A**: Experimental procedure for an example question. **Fig B**: Correlation between median estimate and correct answer for general knowledge VS numerosity questions and for very large VS moderately large quantities. **Fig C**: Significance analysis of the differences between slopes in all panels of Fig 1 as well as of these slopes being lower than 1. **Fig D**: Narrowing of the distributions of estimates after social information sharing in Fig 2 and analysis of its significance. **Fig E**: Probability density function (PDF) of personal estimates $X_p$ for all conditions combined. **Fig F**: Probability density function (PDF) of the fraction of instances with $S = 0$ for each participant and each question. **Fig G**: Significance analysis of the differences in $P_g$, $m_g$ and $\sigma_g$ between treatments in Fig 4. **Fig H**: $\langle S \rangle$ against $\tau$ and $\langle \sigma \rangle$ for the model without similarity effect. **Fig I**: $\langle S \rangle$ against $\tau$, when $D < 0$ and $D > 0$, for the model without similarity effect. **Fig J**: Collective and individual accuracy against $\tau$ for the model without similarity effect. **Fig K**: $\langle S \rangle$ against $\tau$ and $\langle \sigma \rangle$ for the model without asymmetry effect. **Fig L**: $\langle S \rangle$ against $\tau$, when $D < 0$ and $D > 0$, for the model without asymmetry effect. **Fig M**: Collective and individual accuracy against $\tau$ for the model without asymmetry effect. **Fig N**: Significance analysis of the

improvements in collective and individual accuracy in Fig 12. **Fig O**: Significance analysis of the difference in improvement in collective accuracy between treatments in Fig 12. **Fig P**: Significance analysis of the difference in improvement in individual accuracy between treatments in Fig 12. **Fig Q**: Significance analysis of the improvement in individual accuracy in Fig 13. **Fig R**: Collective accuracy against $\tau$ when $D < 0$ and when $D > 0$. **Fig S**: Significance analysis of the improvement in individual accuracy in Fig 14. **Fig T**: Collective accuracy against $\tau$ when $S$ is below and above Median($S$). **Fig U**: Collective and individual accuracy against $\tau$ in the Shifted-Median treatment compared to a simple recalibration of initial estimates. **Fig V**: Collective accuracy against $\tau$ for moderately large and very large quantities. **Fig W**: Individual accuracy against $\tau$ for moderately large and very large quantities. **Table A**: Distribution of cases when the social information provided to an individual was higher ($D > 0$) or lower ($D < 0$) than their personal estimate **Table B**: Goodness-of-Fit and relative error between the data and the model.
(PDF)

## Acknowledgments

We are grateful to Felix Lappe for programming the experiment, and thank Alan Tump, Lucienne Eweleit, Klaus Reinhold, and Oliver Krüger for their support in the organization of our study. We are grateful to the ARC research group for their constructive feedback.

## Author Contributions

**Conceptualization:** Bertrand Jayles, Ralf H. J. M. Kurvers.

**Data curation:** Bertrand Jayles, Ralf H. J. M. Kurvers.

**Formal analysis:** Bertrand Jayles, Clément Sire.

**Funding acquisition:** Ralf H. J. M. Kurvers.

**Investigation:** Bertrand Jayles, Clément Sire, Ralf H. J. M. Kurvers.

**Methodology:** Bertrand Jayles, Clément Sire, Ralf H. J. M. Kurvers.

**Project administration:** Bertrand Jayles, Ralf H. J. M. Kurvers.

**Resources:** Bertrand Jayles, Ralf H. J. M. Kurvers.

**Software:** Bertrand Jayles.

**Supervision:** Clément Sire, Ralf H. J. M. Kurvers.

**Validation:** Bertrand Jayles, Clément Sire, Ralf H. J. M. Kurvers.

**Visualization:** Bertrand Jayles, Clément Sire.

**Writing – original draft:** Bertrand Jayles.

**Writing – review & editing:** Bertrand Jayles, Clément Sire, Ralf H. J. M. Kurvers.

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
