## [Decision Letter · Decision Letter 0]

1 May 2020

Dear Dr. Jayles,

Thank you for your submission, "Debiasing the crowd: selectively exchanging social information improves collective decision-making," for consideration at PLOS Computational Biology. The manuscript provides a thought-provoking perspective on mechanisms of information sharing that improve outcomes in collective decision-making systems, particularly for the case of aggregating judgment among a population of individuals prone to estimation bias. It occurred to me that the methodologies and target application used here could potentially resonate with a broad audience cutting across multiple disciplines, which would make PLOS Computational Biology an excellent venue. Consequently, I chose to sample the judgment of a diverse group of reviewers -- with no two reviewers from the same community of literature -- that would be representative of what I think would be the target audience for an eventually published paper. We were fortunate to have so many of this selection of reviewers agree to review the paper, which should be a good sign to the authors that the subject is of broad interest.

Although the reviewers were diverse and independent, their recommendations were consistent with each other (and relatively easy to aggregate) and also in agreement with my personal feeling toward this manuscript. In its current form, I cannot recommend this manuscript for acceptance in PLOS Computational Biology. The collection of reviewers has outlined a wide range of items that need to be clarified, literature that should be referenced (and terms that should be reconsidered), and important questions to be answered. I do not feel like the critiques from the reviewers pose insurmountable challenges, and so I would like to invite you to submit a major revision of your manuscript that addresses the feedback of the reviewers. I want to emphasize (and caution you) that this major revision is not a formality; a major revision will again be reviewed based on its merits, and the ultimate decision for that revision is not clear at this stage.

It is my feeling that most of the necessary revisions can be made by re-structuring the narrative around these empirical results and better contextualizing these results within the background of the literature highlighted by the five reviewers. In this exercise, I hope that you will also take an opportunity to evaluate whether there are additional links that may have been missed beyond the several examples brought up by these reviewers. For example, your manuscript's results about the effect of group size might also be compared and contrasted with Condorcet's jury theorem or other work describing how many voters are necessary to come to a correct or accurate result. As covered by the reviewers, you should also be careful about using loaded language ("herd") and oversimplifications (like saying that no prior information is necessary for the correction parameter even though the correction parameter was estimated from prior information (albeit from a different group)).

Overall, I do not feel like you should be discouraged by this decision. Clearly, there is broad interest in your submitted results, and we are all looking forward to your major revision to see how you have addressed the significant concerns brought up by a clearly very interested group of referees.

Best wishes to you --

Theodore (Ted) P. Pavlic

Guest Editor, PLOS Computational Biology

We cannot make any decision about publication until we have seen the revised manuscript and your response to the reviewers' comments. Your revised manuscript is also likely to be sent to reviewers for further evaluation.

Sincerely,

Theodore Paul Pavlic

Guest Editor

PLOS Computational Biology

Natalia Komarova

Deputy Editor

PLOS Computational Biology

Thank you for your submission, "Debiasing the crowd: selectively exchanging social information improves collective decision-making," for consideration at PLOS Computational Biology. The manuscript provides a thought-provoking perspective on mechanisms of information sharing that improve outcomes in collective decision-making systems, particularly for the case of aggregating judgment among a population of individuals prone to estimation bias. It occurred to me that the methodologies and target application used here could potentially resonate with a broad audience cutting across multiple disciplines, which would make PLOS Computational Biology an excellent venue. Consequently, I chose to sample the judgment of a diverse group of reviewers -- with no two reviewers from the same community of literature -- that would be representative of what I think would be the target audience for an eventually published paper. We were fortunate to have so many of this selection of reviewers agree to review the paper, which should be a good sign to the authors that the subject is of broad interest.

Although the reviewers were diverse and independent, their recommendations were consistent with each other (and relatively easy to aggregate) and also in agreement with my personal feeling toward this manuscript. In its current form, I cannot recommend this manuscript for acceptance in PLOS Computational Biology. The collection of reviewers has outlined a wide range of items that need to be clarified, literature that should be referenced (and terms that should be reconsidered), and important questions to be answered. I do not feel like the critiques from the reviewers pose insurmountable challenges, and so I would like to invite you to submit a major revision of your manuscript that addresses the feedback of the reviewers. I want to emphasize (and caution you) that this major revision is not a formality; a major revision will again be reviewed based on its merits, and the ultimate decision for that revision is not clear at this stage.

It is my feeling that most of the necessary revisions can be made by re-structuring the narrative around these empirical results and better contextualizing these results within the background of the literature highlighted by the five reviewers. In this exercise, I hope that you will also take an opportunity to evaluate whether there are additional links that may have been missed beyond the several examples brought up by these reviewers. For example, your manuscript's results about the effect of group size might also be compared and contrasted with Condorcet's jury theorem or other work describing how many voters are necessary to come to a correct or accurate result. As covered by the reviewers, you should also be careful about using loaded language ("herd") and oversimplifications (like saying that no prior information is necessary for the correction parameter even though the correction parameter was estimated from prior information (albeit from a different group)).

Overall, I do not feel like you should be discouraged by this decision. Clearly, there is broad interest in your submitted results, and we are all looking forward to your major revision to see how you have addressed the significant concerns brought up by a clearly very interested group of referees.

Best wishes to you --

Theodore (Ted) P. Pavlic

Guest Editor, PLOS Computational Biology

Reviewer's Responses to Questions

**Comments to the Authors:**

Reviewer #1: Jayles and Kurvers investigate how information exchange between group members affects the accuracy of both individual and collective estimations. In particular, they investigate the effect of provision of varying quantities of others’ estimates, and propose a new framework for information exchange based on the shifted-median of previous estimates that improves collective and individual accuracy compared to simply providing estimates from other individuals at random. An agent-based model is used to explore the mechanisms behind this improvement and the dynamics of individual estimate changes.

The central finding in this manuscript is that by ‘leveraging prior knowledge’ about a common underestimation bias, information exchange can be structured to improve collective and individual accuracy. Essentially what this means is that, since we know that people will typically underestimate quantities by a certain proportion, and that they will move towards other estimations they are given, their accuracy can be improved by providing an estimate that is (statistically speaking) likely to be a slight overestimation of the truth.

The study appears well-conducted, and the combination of empirical and modelling work is well constructed. However, as it stands, I do not find the central finding sufficiently significant for publication in PLoS Computational Biology. I would justify this as follows:

1. Although the authors contend that their shifted-median method does not rely on recourse to the truth, this is only true in the sense that the answer to one specific question is unknown. It relies on strong statistical regularities in the relation between individual estimates and the truth. This is what is meant by ‘leveraging prior knowledge about this bias’ in the abstract.

2. Where these regularities apply, it would be more straightforward to simply adjust all individual estimates or the collective estimate (however obtained) directly, rather than by the contrived mechanism of providing individuals with estimates from specially chosen other individuals. There is no sense here that the selective exchange of social information could be generated endogenously from within the group, but instead it is imposed by an external agent. This same external agent could instead manipulate either individual or collective estimates directly.

3. Where these regularities do not apply there is no reason to think that this method would give improved estimations (and could even make them worse). The authors’ own introduction reveals that human estimation and decision making is prone to many contradictory biases (e.g. pessimism and optimism, L43-44). It is unlikely that one could reliably know in advance whether the specific context lends itself to underestimation (though if one could, see point 2 above)

To make a stronger case for the relevance of these results, this manuscript therefore needs a convincing motivation for:

1. The underestimation bias being widespread, important and reliably present, or identifiable in advance (so that the method works and it is known that it will work)

2. Why it is important to affect individuals estimations via the provision of selectively chosen estimations from others, rather than either directly manipulating the original estimates or simply providing alternative information to estimators (e.g. “experience suggests you are likely to be underestimating, consider raising your estimate”).

I would suggest that the authors carefully consider whether this crucial point can be sufficiently motivated before revising the manuscript.

Minor points:

1. It would be interesting to compare the accuracy of the mean as both an individual and collective measure as well as the median. The authors contend that the median of group estimates are more reliable (L132): they may certainly be more robust/less variable for small samples, but it would be useful to see if the stronger effect on the mean of the rare large estimates would counteract the underestimation bias. It would also be good to see if the method used here works for estimation of quantities that are not strictly positive.

2. The modelling work in the manuscript (largely explained in SI) is detailed and shows a good progression of models to explain features of the empirical data. A further suggestion would be to consider how each effect used in the agent-based models can be justified in terms of rational or adaptive behaviour. At the very least, there is an established statistical literature on information integration that could inform the dependence on dispersion of social information.

3. Participants were motivated to be accurate by relatively small financial reward differences based on categories of accuracy. It would be interesting to consider and discuss whether the specific structure of this reward influences the types of estimate received. For example, what if the occasional very large error had a greater or lesser effect on the final reward? Or rewards based on accuracy relative to other participants rather than solely individual accuracy?

4. L44: How can biases be individually rational? I think this requires an example considering that it is unintuitive

5. I felt the manuscript could be clearer in places by using more straightforward English. For example, replacing ‘leveraging’ with ‘using’ and ‘potentialities’ with ‘potential’. This is not to criticise the general standard of English, which is otherwise high.

Reviewer #2: I will state first that I am not in the area of social estimation, and my expertise is more in sequential design of experiments and Bayesian Optimization, which are techniques that could have an important role in the work presented by the authors.

The paper looks into the problem of underestimation bias in a setting with a group of people is asked to provide an estimate and subsequently is provided with several estimations from the remainder of the group (a subset). The authors use three mechanism to select how to exchange information: a random mechanism, a median driven mechanism, and a shifted median exchange, which considers the overestimation bias and compensates for it using a factor γ. The authors proceed optimizing this factor and identify three modeling factors to be considered to study the impact of the amount of exchanged information onto the individual accuracy and collective accuracies. These aspects are herd, asymmetry, and similarity.

The paper was an incredibly interesting read, I believe the authors make a good case on the contribution of their work compared with the available literature, which appear to be herd, asymmetry and similarity. I can appreciate the modeling value in this. However, there are some aspects which I believe the authors should clarify to make the paper contribution clearer and, possibly, stronger.

**What is the effect of the specific population being chosen? Are there aspects of the model that can indicate different outcomes in terms of the accuracy, based on the population? Can we have populations that are over and underestimating?

**It is unclear whether the theory being developed in this paper may apply to non quantitative cases. The authors refer in multiple places to recommendation systems, but more details should be provided on how the very idea of underestimation bias that the authors use to define the coefficient γ, can be extended to cases where no quantitative measures are available.

Detailed comments are attached.

Reviewer #3: This is an interesting paper reporting a model and an empirical study of collective judgment. The study seeks to understand the effects of information sharing in groups and in particular the effects of the amount of information shared and the selection process of which pieces of information are shares. A particular point of interest from the authors’ perspective is how well can a particular process (which they label “shifted median”) designed to counter natural individual judgment “biases” improve the quality of the judgments.

I like the topic and the approach. I think the experiment is well designed and, for the most part, it is well analyzed and clearly reported, but, in my view, this version is not ready for publication. Many of my reservations are related to the writing and presentation style which is imprecise and involves some over generalizations and is, occasionally, sloppy. I will list many of these instances in the order I spotted them in the manuscript, and not necessarily in terms of their importance or severity.

• There is a basic distinction in the literature between judgments and decisions. Decisions involve choices or valuations (usually of competing options) and involve consequences of these actions (often, but not always, monetary). For example – should I invest in A or B? Should I take medication X or Y? How much should I pay for this car, apartment, dress, etc.? Judgments are, as the name indicates subjective estimates of quantities, frequencies, probabilities, etc. that carry no such consequences. This paper is all about judgments and judgment biases, but the authors often refer incorrectly to decisions. This should be corrected throughout the ms.

• The authors refer to a lot of biases without ever properly defining what they mean. In some sense, without proper contextualization, any empirical regularity can be labeled a bias. In the classical work by Tversky and Kahneman biases are defined with reference to a normative model (probability theory) that dictates how judges should act in various circumstances (and even this approach is subject to criticism as in Costello and Watts, 2014), but in many of the cases listed in the paper, I am unsure why things are labeled biases. Many of them (e.g., Optimism, Pessimism) can be explained by other simpler accounts that are totally “unbiased”. This needs to be clearly explained.

• Is there a human tendency to underestimate quantities? I don’t think so! I think that it is fair to say that that human judgment is regressive and people tend to over (under) estimate low (high) quantities (see, for example reference [29] in the paper). This paper focuses primarily on “large” quantities (see the list in appendix), but fails to state this explicitly (exception line 129) and systematically, and creates the false impression that this is a more general pattern. This needs to be corrected throughout.

• There are multiple references to “human tendencies” (see for example line 22 – 23 in the Abstract). I think every one of these (over?) generalizations should be accompanied by some references to back it up.

• I am puzzled by the use of the term “exchange of information”. Every definition I am aware of, stresses the bi–directionality of any exchange, but in this context people are only receiving information from others and they can revise / adjust / refine their judgements in light of this new information, but they don’t offer anything in return, so there is no “exchange”. It is true, that every subject’s judgments are presented to the others in the group, but it is not clear to me that they know this and that there is any reciprocal thinking involved here. So, I would replace the term exchange with one that describes more accurately the setup.

• I think that the three presentation formats are not presented clearly enough, and I think it is worth explaining, what took me a while to recognize, namely that the Median differs from Random simply because it eliminates extreme values (essentially, trimming) and presents only the X/11 (X = 1,3,..11) central values of the distribution, and that Shifted Median presents the same values, but after a systematic shift.

• The statement on lines 96 – 97 is mathematically wrong (or, maybe not clearly stated): It easy to show many cases where the expected Random choice is closer to the truth than the Median. Example: Truth = 10; Assume a group of 6 people such that the 5 potential estimates are 1,2,3,4 and 9 (distances from truth = 9,8,7,6 and 1, respectively). If we choose k = 3 the median selects the estimates 2, 3 and 4 with a mean (and median) distance of 7 from Truth. But, if you consider all 10 (equally likely) different ways to choose 3 of the 5 you get Mean(10 Medians) = 7 and Mean(10 Means) = 6.2! And, if you choose k = 1, the median selects 3 with a mean (and median) distance of 7 from Truth. Under a random choice the mean distance to the truth is, again 6.2! Please clarify / correct.

• In forecasting there is a small literature on re-calibrating probabilities in aggregation (see papers by Baron et al and Turner et al). The shifted median is another, simpler, instance of re-calibration with a twist. In forecasting the transformation is done mechanically and externally after the estimation process. Here the judges are exposed to the re-calibrated judgments of their partners. This brings up an intriguing question. If one was to take the estimates from the Median condition and apply the same shifting transformation, how would these recalibrated aggregates compare to those obtain in the Shifted Median condition? Clearly it is easier to recalibrate things statistically / mechanically, but is it also better?

• I did not fully understand the difference between the collective and individual accuracy measure and I was frustrated by the insufficient and inadequate discussion. I assume that the authors calculate an individual measure for each of the 216 people based on their 36 judgments and that they calculate the collective accuracy for each of the 18 groups based on the group’s (12 members X 36 items =) 432 judgments. That is the way I would have done this but I am not sure this is what was done. I would like to see a better and clearer description.

• Related: On page 6 in the results section, you write “individual accuracy measures how close individual deviations from the truth are to 0 on average,” while technically your individual accuracy measure is an individual’s median accuracy across questions (correct?), not the mean.

• The items (listed in Appendix) are clearly of two types: The majority is based on general knowledge of actual facts such as “What is the population of X?” and a minority ask for a perceptual impression / estimation (e.g., “How many marbles are in the jar”?). It would be nice to present some evidence that the degree of underestimation is similar in the two classes (for example, use different colors for the two in Figure 1) and that the proposed method works equally for both.

• Line 166: when k = (n-1) = 11 one expects identical responses under various condition only in the absence of overweighting of one’s own original judgment (egocentric weighting), which is often seen in the literature (e.g. Yaniv & Kleinberg, 2000) and clearly in the present data (note that most values of S < 0.5).

• To make sense of the asymmetry effect and Figure 3, we need to know what is the distribution of cases where the weight of the social information is <, approximately = or > than one’s own.

• Related: Could a possible explanation for the asymmetric effect be that people, in fact, have some intuition that they tend to underestimate large quantities? Seeing others provide larger estimates than their initial belief may be a sort of cognitive permission to be more liberal with their beliefs about large quantities. While seeing smaller estimates may be seen as typical and expected.

Thinking about the results at a more general level:

Consider the improvements to collective accuracy as predicted by the model in the dashed lines of Figure 2a. Why does the model predict that the random selection method will improve collective accuracy more than the median selection method? This is especially puzzling, since this does not seem to reflect the actual differences between these groups, where it appears the random selection method was more linearly related to the number of estimates exchanged; while the median selection method was flatter across the number of estimates exchanged.

A possible intuition for this result might have to do with how the distance effect parameter was treated. It appears that the relationship between distance and belief updates was treated as linear and increasing, but this is not necessarily the universally observed expectation based on the social persuasion literature. For example, Whittaker (1963) found a curvilinear relationship between distance and belief updates. Other studies have found similar results (Fink, Kaplowitz, & Bauer, 1983; Laroche, 1977; Yaniv & Milyavsky, 2007) and Allahverdyan & Galstyan (2014) proposed a formal model incorporating this effect.

Looking at figure S6d, it does look like a curvilinear parameter might fit the data better than the linear one proposed. When combined with your model’s asymmetry effect, it seems possible that providing random estimates could lead to this somewhat odd result in Figure 2a. Random estimates are more likely to be extreme than median estimates (which are by definition the least extreme available). However, the asymmetry effect parameter means extreme estimates that are lower than the judges’ initial estimates get discounted, while ones that are higher do not. This could induce a somewhat artificial correction for the underestimation bias that isn’t present in the observed data. Treating the distance parameter as non-linear could potentially correct this and may be worth trying.

A somewhat related question is whether there were any effects of bracketing (see Herzog & Hertwig, 2009; Larrick & Soll, 2006; Soll & Mannes, 2011). The authors provide a formalization for how individuals incorporate the social information based on its geometric mean and standard deviation, but do not discuss whether people treat this information differently in cases where the estimates bracket their beliefs (whether their beliefs are within the bounds of the different estimates they receive) or do not. Normatively, in cases where people believe the estimates they receive bracket the truth, they should be more inclined to average and weight that advice fairly heavily; while in cases where the estimates they receive do not bracket the truth they should not (though this normative principle is not always observed in behavior). One could argue that estimates which bracket a judge’s initial estimate could be considered a bracket around their a priori belief about the truth, which would make such brackets qualitatively different from ones that do not. This could also have implications for comparing the random and median conditions. Especially when the number of estimates received is small, the diversity of random estimates may be more likely to bracket a judge’s initial beliefs; while the more homogenous median estimates may be less likely to.

Thank you for the opportunity to review this thought provoking paper

David Budescu

References

Allahverdyan, A. E., & Galstyan, A. (2014). Opinion dynamics with confirmation bias. PloS One, 9(7).

Baron, J., Mellers, B.A. Tetlock, P.E. Stone, E., Ungar, L.H. (2014) Two Reasons to Make Aggregated Probability Forecasts More Extreme. Decision Analysis 11(2):133-145.

Costello, F. & Watts, P. (2014). Surprisingly rational: Probability theory plus noise explains biases in judgment. Psychological Review, Vol 121(3), J 463-480

Fink, E. L., Kaplowitz, S. A., & Bauer, C. L. (1983). Positional discrepancy, psychological discrepancy, and attitude change: Experimental tests of some mathematical models. Communications Monographs, 50(4), 413–430.

Herzog, S. M., & Hertwig, R. (2009). The wisdom of many in one mind: Improving individual judgments with dialectical bootstrapping. Psychological Science, 20(2), 231–237.

Laroche, M. (1977). A model of attitude change in groups following a persuasive communication: An attempt at formalizing research findings. Behavioral Science, 22(4), 246–257.

Larrick, R. P., & Soll, J. B. (2006). Intuitions about combining opinions: Misappreciation of the averaging principle. Management Science, 52(1), 111–127.

Soll, J. B., & Mannes, A. E. (2011). Judgmental aggregation strategies depend on whether the self is involved. International Journal of Forecasting, 27(1), 81–102.

Turner, B.M., Steyvers, M., Merkle, E.C., Budescu, D.V., & Wallsten, T.S. (2014). Forecast aggregation via recalibration. Machine Learning, 95, 261-289.

Whittaker, J. O. (1963). Opinion change as a function of communication-attitude discrepancy. Psychological Reports, 13(3), 763–772.

Yaniv, I., & Milyavsky, M. (2007). Using advice from multiple sources to revise and improve judgments. Organizational Behavior and Human Decision Processes, 103(1), 104–120.

Yaniv. I. & Kleinberg. E. (2000). Advice Taking in Decision Making: Egocentric Discounting and Reputation Formation. Organizational Behavior and Human Decision Processes, 83(2), 260-281.

Reviewer #4: See attached.

Reviewer #5: In this manuscript, the authors extend previous work, by themselves as well as others in the field, to examine how social influence can affect individual and collective accuracy in estimation tasks. Specifically, here they examine how selecting the how many, and which, estimates to give to participants can improve decision accuracy by counteracting known estimation biases. Among other findings, they find that by providing participants with estimates closer to a modified median can substantially improve collective wisdom. They extend a mechanistic model of social influence incorporating the new phenomena that they identified in this study and find that their model can reproduce, to a large extent, their empirical findings. Furthermore, they use their model to identify an optimal strategy to maximized collective wisdom.

This work is a natural extension of research that has appeared in the past couple of years and makes some important findings by using a clever experimental design that clarifies some outstanding questions related to social influence and the wisdom of crowds. As such, I think that this is an important work that is highly suitable for PLoS Comp Biol. Also, I think that this manuscript is well written and the methods and analyses are sound -- as such, I would recommend publication of this manuscript almost as is. I only have a few minor comments that I hope will improve the clarity of the manuscript:

1. I think the paper Becker et al (2017) Network dynamics of social influence in the wisdom of crowds. PNAS should probably be cited somewhere since it speaks to a lot of the same issues as the present manuscript (how who-influences-who can affect the wisdom of crowds).

2. "our method does not require the a priori knowledge of the truth" (lines 143-144). While I know what the authors mean by this, one could in theory disagree with this statement because the parameterization of their model (i.e., that gamma = 0.9) requires knowledge of some previous truths. However, their method does not require knowledge of the truth for the present estimation task. This could be clarified.

3. line 192: I would argue that the asymmetry effect was described to some extent in reference 44 of this manuscript: Kao et al (2018) Counteracting estimation bias and social influence to improve the wisdom of crowds. J Royal Society Interface (Disclaimer: since I'm one of the authors of that manuscript, I've signed this review in the pursuit of transparency) In Figure 5a and 5c of that paper, we described a similar effect, where estimates larger than the focal individual's were weighted more heavily than estimates that were smaller. However, the empirical trend in the present manuscript is somewhat different, with a stronger effect size. In any case, it may be useful to point this out somewhere, to show that this asymmetry effect may be robust and widespread (although I refrain from pushing this point too strongly since it is a paper that I'm a co-author on).

Sincerely,

Albert Kao

**Have all data underlying the figures and results presented in the manuscript been provided?**

Reviewer #1: No: Current data availability statement is vague. The authors should specifiy where/how data will be made available on acceptance, and ideally provide data to reviewers for review.

Reviewer #2: Yes

Reviewer #3: None

Reviewer #4: Yes

Reviewer #5: Yes

PLOS authors have the option to publish the peer review history of their article (what does this mean?). If published, this will include your full peer review and any attached files.

Reviewer #1: No

Reviewer #2: No

Reviewer #3: No

Reviewer #4: No

Reviewer #5: Yes: Albert B. Kao
---

## [Decision Letter · Decision Letter 1]

16 Sep 2020

Dear Dr. Jayles,

Thank you very much for submitting your manuscript "Debiasing the crowd: how to select social information for improving collective judgments?" for consideration at PLOS Computational Biology.

As with all papers reviewed by the journal, your manuscript was reviewed by members of the editorial board and by several independent reviewers. In light of the reviews (below this email), we would like to invite the resubmission of a significantly-revised version that takes into account the reviewers' comments. Please address the reviewers' comments in your revisions.

We cannot make any decision about publication until we have seen the revised manuscript and your response to the reviewers' comments. Your revised manuscript is also likely to be sent to reviewers for further evaluation.

Sincerely,

Theodore Paul Pavlic

Guest Editor

PLOS Computational Biology

Jason Papin

Editor-in-Chief

PLOS Computational Biology

Natalia Komarova

Deputy Editor

PLOS Computational Biology

Reviewer's Responses to Questions

**Comments to the Authors:**

Reviewer #1: I am grateful to the authors for providing a detailed response to my previous review, and note that they have made substantial revisions to address reviewer comments. I would also note that having read the other reviews, I find that I am somewhat out of step with the other reviewers in my assessment of the interest/importance of the manuscript. I have no technical objections to the work in the manuscript, but below I would like to justify why I continue to believe the results are below the importance threshold for PLoS CB.

In their response to reviews the authors have repositioned their study as a test of how individual estimates of unknown quantities can be improved in social contexts. They justifiably note that they did not seek to find out how to best combine independent estimates to obtain collective accuracy. However, the text of the manuscript does not concord with the purported focus on improving individual judgements. The title of the manuscript points to the goal of improving collective judgements as the primary goal, a pattern that is repeated through the abstract (L8-9, L19, L26) and introduction (L55-56, L57-58, L69). In the results, on the occasion that individual and collective improvements are at odds, collective improvement is favoured (albeit at only minor cost to individual improvement, L 272-273). While the innovative experimental treatment (shifted-median) offers clear collective improvements, the benefit at the individual level is far from clear when compared to more standard information exchange (either random or median, Fig 2.). Only in the discussion is the revised goal of individual improvement placed at the fore (L361-362), but this is a departure from the large majority of the material that comes before.

As a result of the above, I don’t think the study can be evaluated without putting the collective efficacy of the methodology at the forefront; collective improvements are not presented as a happy consequence of individual improvement, but stand out as the most noted and noteworthy results. In this light, I consider that my previous criticism of the manuscript broadly remains: as long as the sharing of carefully chosen social information has to be coordinated by a central agent that knows the full distribution of initial estimates, it does not offer a efficient alternative method for ‘improving collective judgements’ as offered by the title of the manuscript. Since it is well-established that individuals will adjust their estimates towards those they see from others (on average), it is unsurprising that carefully choosing what they see so as to make it (statistically) more accurate will produce more accurate estimates overall. If this is coordinated centrally it is much the same *as if* individual estimates had been combined in an optimal way. If a mechanism could be designed to facilitate this biased information sharing operate endogenously (thus removing the central organising agent) then this criticism would fall away, but I cannot suggest any way to achieve this, especially as the central coordinator must also decide in advance if this estimation problem is one that belongs to the domain of typical under- or over-estimation.

Reviewer #2: I am happy with the responses and revisions of the authors and I think the papers is publishable in the current form.

Reviewer #3: This is an interesting paper reporting a model and an empirical study of collective judgment. The study seeks to understand the effects of information sharing in groups and in particular the effects of the amount of information shared and the selection process of which pieces of information are shared. A particular point of interest from the authors’ perspective is how well can a particular process (“shifted median”) designed to counter natural individual judgment biases improve the quality of the judgments. I like the topic and the approach. The experiment is well designed and, for the most part, it is well analyzed and clearly reported.

The authors addressed seriously most of my reservations and this version is much better. I have few outstanding issues/questions:

1. In clarifying the underestimation effect, the authors state “The underestimation bias is a widely documented human tendency to underestimate large quantities (typically larger than 100) in estimation tasks”. Their explanation and qualification with regard to the underestimation effect is improved. However, this example of “typically larger than 100” seems a bit odd. Isn’t there some degree of scale dependence? Would one expect the bias in describing something as a matter of 120 seconds, but not 2 minutes? The authors proceed to provide domain examples where the underestimation effect could be expected. It strikes me that it might be preferable to expand on one of these examples a bit rather than use a sort of arbitrary, scale-free pseudo-criterion like 100. What it is it about population estimates that makes it susceptible to the bias? Can you perhaps describe, or provide an example of, the distribution and its key properties? It also seems like this would serve as good motivation for the log transformations performed (though this is more clearly explicated in the revised version, which I appreciate).

2. The definitions of collective and individual accuracy are much improved. One further suggestion is that they could perhaps use slightly more qualitative motivation, more like what was provided in their response letter: “individual accuracy is a measure of the distance of an individual’s estimates from the truth, and collective accuracy is a measure of how far the central tendency of the estimates of the group is from the truth.” I think this sentence is helpful in what is a critical and potentially confusing distinction, and it is worth including something to this effect in the manuscript.

3. I am puzzled by the prediction regarding the random condition. The idea that people would be insensitive to the number of pieces of advice presented to them is counterintuitive (after all, everyone can do exactly what the authors are doing in the median condition, namely reject/ignore extreme values), and is inconsistent with empirical evidence about the way people aggregate information from multiple sources (e.g., Budescu & Rantilla ,2000; Budescu, Rantilla, Yu & Karelitz, 2003; Budescu & Yu, 2007).

4. Line 296: The tradeoff between bias and other factors in WoC is analyzed in Davis-Stober, Budescu, Dana & Broomell, 2014).

5. The authors seek to identify the “optimal” adjustment and refer to how close the shift is to the “true” value. I don’t know how seriously to take the notion that there is a “true” value. The target they use is based on the degree of underestimation observed in a study using a particular set of items / questions and I am not sure that it would replicate with different items (imagine asking people to estimate distances to various planets). I think some measure of caution and qualification is needed.

6. The one point I remain somewhat unpersuaded about is the explanation regarding the empirical results in figure 2a. Here is the question and response from the response letter:

16. This is especially puzzling, since this does not seem to reflect the actual differences between these groups, where it appears the random selection method was more linearly related to the number of estimates exchanged; while the median selection method was flatter across the number of estimates exchanged.

We agree that Fig. 2a gives the visual impression that collective improvement in the random treatment increases more linearly than in the median treatment, mostly because the blue point at tau = 1 is way higher than predicted by the model (and expected by us). Apart from this single point, both treatments increase, as predicted, linearly and it is likely that this single point deviated too much from its expected value due to noise, as often happens with limited samples. Though we tested 216 participants over 36 questions, our sample size per unique treatment combination is still limited. So, we believe we need to treat each single point with caution, but can have much more confidence in the general patterns.

While the non-linearity intuition may not have borne out, it still seems like this empirical pattern is more than just a single data point. While τ = 1 may be the most extreme deviation from the model prediction, it looks like the error bars for almost every point in the median condition almost entirely overlap. I’m not sure writing it off as sample size noise in a single treatment condition is fully justified. It is a possible explanation, but beyond the (tested) suggestion about non-linearity, isn’t a more straightforward alternative simply that in the median condition most of the benefit of social information comes from that first piece of information? The Wisdom-of-Crowds theory tells us that the median response has the expectation of being the most accurate single piece of social information available. Each subsequent piece of information (by definition not the median) is expected to be less accurate, so while the similarity and herding effects may tell us that more advice may increase the magnitude of the belief update, the actual expected value of the advice cannot improve. There is a necessary tradeoff between the impact and accuracy of extra information in this condition. In other words, there is possible theoretical reason to expect that, in the median condition, increasing τ should have less added benefit.

The authors suggestion of caution is warranted. However, the full extent the authors pay to this is to parenthetically note “(though it is unexpectedly high for τ = 1).” I’m not sure this is a sufficient treatment of this result, given how easy it is to come up with intuitive explanations. At the very least, further remarks in the context of replication seem warranted, and it may be possible to test this possible explanation on your data as well with minor modifications to your model.

Picky

• Footnote 2: There is no MLE for the “center” of a distribution; there are MLEs for well-defined statistical parameters (mean, median, mode, etc.)

• Line 197: Add (0 ≤ S ≤ 1).

Thank you for the opportunity to review this thought-provoking paper

David Budescu

References

Budescu. D.V. & Rantilla, A.K. (2000). Confidence in aggregation of expert opinions. Acta Psychologica, 104, 371-398.

Budescu, D.V., Rantilla, A.K, Yu, H., & Karelitz. T.M. (2003). The effects of asymmetry among advisors on the aggregation of their opinions. Organizational Behavior and Human Decision Processes, 90, 178 – 194.

Budescu, D.V., & Yu, HY. (2007). Aggregation of opinions based on correlated cues and advisors. Journal of Behavioral Decision Making, 20, 153-177.

Davis-Stober, C.P., Budescu, D.V., Dana, J., & Broomell, S.B. (2014). When is a crowd wise? Decision, 1, 79-101.

Reviewer #4: Please see attachment.

Reviewer #5: I appreciate the authors' efforts to respond to the comments and criticisms of _five_reviewers. I had only minor comments in the last round (reviewer 5), and unsurprisingly the authors have sufficiently addressed them in the current revision. However, I did take the time to read through all of the other reviewers' comments, and the authors' responses to them. I have some comments to the authors' responses to these comments, as well as a couple of new comments to the current version.

1. Some of the other reviewers (reviewer 1, comment 2; reviewer 2, comment 3; reviewer 3, comment 8) have mentioned the broader context of interventions that researchers could use to improve collective accuracy. To all of these comments, the authors responded that they seek to improve individual accuracy, rather than collective accuracy, thereby differentiating their paper from others. However, I disagree with this characterization of their paper. Collective accuracy, in addition to individual accuracy, comprises a major part of their results (e.g., Figure 2a, Figure 5). Furthermore, collective accuracy is mentioned multiple times, for example in their abstract and introduction (e.g., "cognitive biases... can impair the quality of collective judgments and decisions", "our restructuring of social interactions... substantially boosted collective accuracy", biases at the individual level can have negative consequences at the collective level").

Therefore, dismissing these reviewers' comments as irrelevant seems unjustified. I think that the authors could go in two directions:

(a) Substantially edit their text and results to really focus on improving individual accuracy, as they claim to solely do.

(b) Address some of the comments made by the reviewers, specifically, how their methodology compares with the universe of alternative methods to improve collective accuracy.

To add to this, if the authors are OK with manipulating what social information individuals have access to, then why not just generate fake social information, if the goal is simply to maximize individual/collective accuracy? If the authors now know the rules that individuals on average follow, then they could construct a set of completely fake social information that should push the individual exactly towards the correct answer (on average). Selecting only from the set of real social information seems to be a limitation, especially when group size is small. Moving towards fake social information would provide much more flexibility on the part of researchers. If this is true, then perhaps the "optimising collective and individual improvements" section could be modified with this more general intervention.

Or perhaps there is an ethical reason to not completely make up social information? If so, it's not clear to me why carefully selecting what social information an individual sees is so different from just making up the information. Perhaps there could be some discussion about the ethical considerations in these kinds of interventions.

2. Are the statements made in lines 180-186 backed up by some kind of statistical or quantitative analysis, or just mad by "visual inspection"? Particularly, the statement that "individual improvement also increases with tau in the Random treatment" seems dubious, as is, to a lesser extent, the statement that "individual improvement is generally higher in the Median and Shifted-median treatments than in the Random treatment."

3. The slopes of the linear regression lines in Figure 1 could be printed in the figure panels themselves, which I think would improve the clarity of this figure.

**Have all data underlying the figures and results presented in the manuscript been provided?**

Reviewer #1: Yes

Reviewer #2: None

Reviewer #3: None

Reviewer #4: Yes

Reviewer #5: Yes

PLOS authors have the option to publish the peer review history of their article (what does this mean?). If published, this will include your full peer review and any attached files.

Reviewer #1: No

Reviewer #2: **Yes: **Giulia Pedrielli

Reviewer #3: No

Reviewer #4: No

Reviewer #5: **Yes: **Albert B. Kao
---

## [Decision Letter · Decision Letter 2]

22 Jul 2021

Dear Dr. Jayles,

Thank you very much for submitting your manuscript "Debiasing the crowd: how to select social information to improve judgment accuracy?" for consideration at PLOS Computational Biology. I apologize for a delay in the processing of this manuscript. You will find the referee comments below, along with the analysis of the Guest Editor. It is especially important that you address the comments relating to the statistical procedures implemented in your manuscript, as indicated by two of the referees and the Guest Editor. 

As with all papers reviewed by the journal, your manuscript was reviewed by members of the editorial board and by several independent reviewers. The reviewers appreciated the attention to an important topic. Based on the reviews, we are likely to accept this manuscript for publication, providing that you modify the manuscript according to the review recommendations.

Sincerely,

Natalia L. Komarova

Deputy Editor

PLOS Computational Biology

[LINK]

Comments by the Guest Editor:

1. The contribution of this paper relative to the Interface article from Jayes et al. (2020) [21] and the preprint from Jayes et al. (2020) [48]. As discussed by Reviewer 4, both of these papers have significant overlap with the methods and aims of this paper. Furthermore, despite the overlap in authorship among the papers, they are referenced in the current manuscript as if they are independent support for the foundational arguments used in this manuscript. Instead, it appears like these papers were work done in parallel (potentially as part of one larger project), and this needs to be made more explicit. See detailed comments by Reviewer 4.

2. Several reviewers (explicit statements by Reviewers 4 and 5) have pointed out that the empirical arguments made by the authors lack statistical rigor. Some examples:

2.a) The value of gamma has been taken from a linear regression, but there are no data about the (adjusted) R^2 for this regression nor even a p-value indicating that this value of 0.9 is significantly different from 1.0. The value of gamma has been said to be visually similar across three different data sets, but no statistical test was used to justify that these three data sets are likely to have the same value (and that that value is significantly different from 1.0).

2.b) The experimental design makes use of paired before/after data, but the authors do not formally use any paired statistical analyses that would incorporate statistical blocking to account for variance that otherwise confounds the comparison across the different treatment groups.

2.c) The authors claim that the simulation model reproduces the empirical results well, but they do not attempt a formal goodness-of-fit analysis.

2.d) In general, the authors need to consider formal statistical analyses when making inferences about any empirical data -- especially when small numbers of replications are used. Visual arguments (or even arguments focused only on comparing means) are not convincing. Reviewer 5 points out that error bars represent a single standard deviation, which is a significant under estimation of confidence intervals. If we visually approximate confidence intervals by doubling the current error bars, the resulting bars show that different treatment conditions show a large overlap in response. Really showing that there is an effect requires a statistical test in this case. Even there is a significant effect demonstrated, the inferred effect size should be discussed.

In addition, some details of the experimental design pose some confusion. As pointed out by Reviewer 4, the authors state at one point that the FIRST choice of an individual seems to be influenced by the number of estimates shown to that individual. However, the experimental design should be such that the first choice by an individual occurs before any other choices are displayed. This may indicate that some rephrasing is needed, or the experimental design needs to be clarified.

3.) As discussed by Reviewer 3, the real value of the model over what is already shown empirically is not clear. As the authors have used the model, it should act as a lens helping to bring clear focus on which of several different hypothetical drivers are likely responsible for differences seen in the empirical data. However, the model does not currently complement the empirical data to provide clarity; it seems to supplement the empirical data and possibly just raise more questions. That said, I can understand how the authors might feel that without a computational model (and in light of the comments I will make in point 4 below), there may not be much reason this article would belong in PLOS Computational Biology. If the model really is the strong point of the paper that is what anchors it to this journal, then its contribution needs to be made more clear. If the model is removed from the paper, then I would recommend the authors lean on the relevance of this study to recommender systems (as they have already done). But, in the end, if the paper really becomes an empirical study of human psychology, it may be a better fit for PLOS ONE instead.

4.) The authors have gone to great lengths within the text of the article to focus on how information from the crowd can be used to reduce bias in the individual. Previous comments from reviewers have focused on how this paper is not about collective intelligence so much as leveraging information from an ensemble of other evaluators to help improve outcomes from the next evaluator. Still, the title of the paper starts with, "Debiasing the crowd," which suggests that the paper is about designing information sharing mechanisms to improve group outcomes. That is simply not the focus of this paper. The authors might want to consider an analogy to "control variates", a method employed for variance reduction in Monte Carlo methods. In variance reduction, each experimental replication has a multivariate output -- one (X) with a mean that is trying to be inferred, and another with a known mean (Y, Ybar). Control variates use the demonstrated correlation between the two response variables (cov(X,Y), which can be estimated from the data) in a similar way as the "S" variable described by the authors. In particular, the response variable (Y) with the known mean (Ybar) generates a difference from that known mean (Y-Ybar), and that difference can be scaled (with magnitude related to cov(X,Y)) to directly adjust the observed value of the focal variable (X). The authors seem to be asserting that a similar process goes on within the head of an individual when making the second prediction, and their method leverages this to try to reduce the bias in an individual. So, a more accurate title might be something like:

"Crowd Control: How to select social information to improve individual judgment accuracy"

Personally, I prefer titles that state the main results as opposed to posing questions that the reader is promised to find a an answer to within. With that in mind, I might suggest something like:

"Crowd Control: Reducing individual estimation bias by sharing biased subsets of evaluations from others"

That said, I find that one thing diluting the value of this article is that it appears to be two things at once. On one hand, it attempts to be a scientific article making inferences about how humans use social information. On the other hand, it attempts to be a design article suggesting how recommender systems (or other technologically enabled systems) might reduce intrinsic bias in the choices made by their users. I think the article would be improved if the authors would focus on one (and possibly leave the other as a short set of comments in discussion related to broader impacts). My personal recommendation would be to focus on better illuminating the four effects that relate to how humans make use of data from existing raters. Then the design comments could be left for (brief) discussion. If this was the focus, then the title of the paper would not be built around the idea of an action ("Debiasing" or "Control") but instead would be constructed to communicate novel psychological insights (e.g., "Human numerical estimation errors are highly sensitive to...").

Thank you for your time and efforts on this manuscript. I hope you find these comments to be constructive. Best wishes to you in your efforts to further revise this manuscript, if you choose to do so.

Theodore (Ted) P. Pavlic

Guest Editor

PLOS Computational Biology

Reviewer's Responses to Questions

**Comments to the Authors:**

Reviewer #1: The authors have revised the manuscript significantly to account for concerns of myself and other reviewers. I think the manuscript should now be accepted in its current form.

Reviewer #3: This is the third time I am reading this paper that reports a model and an empirical study of collective judgments. The study seeks to understand the effects of information sharing in groups and in particular the effects of the amount of information shared and the selection process of which pieces of information are shared. A particular point of interest from the authors’ perspective is how well can a particular process (“shifted median”) designed to counter natural individual judgment biases improve the quality of the judgments. I like the topic and the approach. The experiment is well designed and, for the most part, it is well analyzed and clearly reported. The authors addressed seriously my reservations and this version is better. I still have few outstanding issues/questions:

In the previous draft of the paper, I found the examples the authors used for the magnitude of the overestimation bias somewhat arbitrary (they referenced the number 100 at some point). They took my suggestion and, instead, focused on a single example, using a dot estimation task, but I still find this slightly opaque and would prefer it were expanded a bit. Can the authors provide a brief explanation of the design of the task, with particular emphasis perhaps on the range of dots participants were expected to estimate? Just slightly more context would go a long way here I feel.

I continue to be puzzled by the prediction regarding the random condition. The notion that people are insensitive to the number of pieces of advice presented to them is counterintuitive (after all, everyone can do exactly what the authors are doing in the median condition, namely reject/ignore extreme values), and is inconsistent with empirical evidence about the way people aggregate information from multiple sources (e.g., Budescu & Rantilla ,2000; Budescu, Rantilla, Yu & Karelitz, 2003; Budescu & Yu, 2007). The authors point out that their results in the random condition confirm this expectation, but this seems to be a bizarre twist of logic. I was not questioning their results, but asking for the justification for the a-priori prediction that runs counter to (at least, some) data. One can’t defend / justify a prediction, by simply, saying it was right! Do they imply that the prediction was preceded by the results?

I think we had this argument in a previous round, but I don’t think the claim on lines 114-115 is mathematically correct. If all judgments over (or under) estimate the true value (i.e. all errors have the same sign), it is easy to generate distributions where the expected error (i.e. the error of a randomly selected judge) is smaller than the error of the median. This needs to be clarified.

Line 172: What exactly does “reliable” mean here? Please define and clarify whether this is an empirical or a normative / theoretical argument. BTW Han & Budescu (2019) show the superiority of the median over the mean in a bunch of cases, and cite other papers doing so.

Line 196: Provide some data to make the point: “In X of the 18 distributions considered, the MSE (or some other measure) of the second estimate is smaller that the MSE (or whatever other measure you pick) of the first set of estimates”.

The model. On line 324 you outline a two-stage process: In a certain fraction of cases judges stick to their estimates (S=0) and in the other cases they draw S from a properly parametrized Normal distribution. As a psychologist, I am bothered by the lack of differentiation between individual and aggregate level theorizing. Figure 3 is based on all judges and all items combined and the spike of S=0 does not differentiate between the two. My question, as a psychologist, is whether the S=0 represents a subset of judges that stubbornly refuse to be influenced by social influence, a subset of items that are so easy that no one looks at the others’ estimates, or whether this reflects a uniform / universal tendency of all judges to stick to their estimates in a fraction of cases (e.g., when they feel very sure about them.) It is an easy analysis to run (the % of cases where S=0 for every judge and every item) that would help clarify this point.

The general discussion: I wish some of the four mechanisms listed would be qualified to reflect the restrictions imposed by the experiment. For example, Schultze, Rakotoarisoa and Schulz-Hardt (2015) show that the distance effect is not always monotonic, and it is possible to imagine case where people would systematically pay more attention to estimates that are lower than theirs (for example, for very rare and / or very undesirable quantities).

Finally, what was Figure 2, and now Figure 12. The new figure contains more information, but I’m not sure it’s more informative. What is persuasive about this figure is essentially the same thing as before: when τ is small, median shifted social information benefits collective accuracy meaningfully more than random social information or the unshifted median. I think this is the most interesting result in the paper and was also clear in the previous version of this figure. I don’t see how the reference lines for the model predictions are informative / useful, the solid line. This appears to just be the expected pre-advice accuracy estimate across all questions and conditions. How is the model’s prediction, which is based on social information, informative in the absence of social information?

More critically, how can we make a comparison with the empirical results when the model prediction is independent of the empirical results by condition? The pre-SI empirical results are condition specific and don’t correspond particularly well to the reference line, which makes comparison between the post-SI empirical results and model predictions extremely difficult. One way to address this might be to leave the solid line out entirely and condition the post-SI model predictions on the pre-SI empirical results. Without something like this, it is very difficult to interpret how well the model is reproducing the empirical patterns. Perhaps this was a hidden issue in the previous versions of this figure, as I may have been implicitly assuming this was the case. Otherwise, this seems like an apples-to-oranges comparison.

Overall, the results (e.g. Figure 10 in particular) seem persuasive that the model adequately captures patterns in individual behavior with regard to social information, but Figure 12 stands out as being difficult to interpret with regard to how well the model is capturing empirical patterns in collective accuracy.

Thank you for the opportunity to review this thought-provoking paper

David Budescu

Han, Y. & Budescu, D.V. (2019) A universal method for evaluating the quality of aggregators. Judgment and Decision Making, 14, 395-411.

Schultze, T., Rakotoarisoa, A., Schulz-Hardt, S. (2015) Effects of distance between initial estimates and advice on advice utilization. Judgment and Decision Making,

Reviewer #4: Please see PDF attached.

Reviewer #5: I appreciate the authors' work to improve their manuscript, and to address the comments of all of the reviewers. This version of the manuscript looks _significantly_ revised, so I have a few new comments to this version that have to do with the poor statistical treatment of the data.

1. "visual inspection confirms" (l.181) This is a very weak way to compare whether or not two sets of data are statistically different from one another. I suggest that the authors use a more rigorous statistical method to do this comparison.

2. "we find narrower distributions after social information sharing" (l.196) This is really not obvious to me, especially if you ignore the lines (which are 'model simulations') and just look at the datapoints. Again, here a rigorous statistical method to make this claim is needed.

3. "the distributions of Xp (solid lines) are simulated by drawing the Xp from Laplace distributions" (l.198) This is odd. There is a closed-form expression for the PDF of the Laplace distribution, so simulating this distribution is unnecessary. Furthermore, it appears that the authors simulated the distribution once, and plotted the same simulation across all of the panels of figure 2, and all of the stochastic jaggedness is identical across the panels. Why not just plot the exact form of the distribution?

4. "A similar pattern of social influence strength is observed at intermediate values of tau (tau = 3, 5, 7, or 9), where Pg and mg are substantially higher in the Median and Shifted-Median treatments than in the Random treatment. For sigma_g, we observe a higher value in the Random treatment than both other treatments at tau = 3 and 5, but not at higher levels of tau." (l.248-251) This appears to be a poor analysis of the data shown in figure 4. First, we can take the figure at face value. Doing so, we can see that Pg does not appear to be "substantially higher" in the Median and Shifted-Median treatments compared to the Random treatment (the blue and black error bars overlap quite a lot). Similarly, sigma_g does not look higher for the Random treatment compared to the other treatments at tau = 3 (the black error bar overlaps both the blue and red error bars).

Worse, however, is the fact that the error bars represent ONE standard error. If we double the length of the error bars to approximate a 95% confidence interval, we can see that nearly all of the error bars will overlap with one another for most of the figure, rendering the authors statements about the data in the text unsubstantiated by the data. The authors need to address this.

5. "we find that the center of the cusp relationship is located at D = D0 <0" (l. 292) However, in line 302, we find that "visual inspection was used to fix D0." Again, this is a surprisingly poor method for determining this, and the authors should use a statistically rigorous method instead.

**Have the authors made all data and (if applicable) computational code underlying the findings in their manuscript fully available?**

Reviewer #1: Yes

Reviewer #3: Yes

Reviewer #4: None

Reviewer #5: None

PLOS authors have the option to publish the peer review history of their article (what does this mean?). If published, this will include your full peer review and any attached files.

Reviewer #1: No

Reviewer #3: **Yes: **David V Budescu

Reviewer #4: No

Reviewer #5: No

Figure Files:

Data Requirements:

Reproducibility:

References:

---

## [Editor Report · Decision Letter 3]

25 Oct 2021

Dear Dr. Jayles,

We are pleased to inform you that your manuscript 'Crowd control: reducing individual estimation bias by sharing biased social information' has been provisionally accepted for publication in PLOS Computational Biology.

Best regards,

Theodore P. Pavlic

Guest Editor

PLOS Computational Biology

Natalia Komarova

Deputy Editor

PLOS Computational Biology

Thank you for your considerable efforts responding to the questions, comments, and concerns of the reviewers and the editing staff. I believe that the current version of the manuscript is ready for larger scrutiny by the broad PLOS Computational Biology audience and will likely be a very thought provoking contribution.

---

## [Editor Report · Acceptance letter]

8 Nov 2021

PCOMPBIOL-D-20-00065R3 

Crowd control: reducing individual estimation bias by sharing biased social information

Dear Dr Jayles,

I am pleased to inform you that your manuscript has been formally accepted for publication in PLOS Computational Biology. Your manuscript is now with our production department and you will be notified of the publication date in due course.

With kind regards,

Katalin Szabo
